# Computer-aided biochemical programming of synthetic microreactors as diagnostic devices

Alexis Courbet[1,2,*,†,‡] iD, Patrick Amar[1,3] iD, François Fages[4], Eric Renard[2,5] & Franck Molina[1]

## Abstract

Biological systems have evolved efficient sensing and decision-making mechanisms to maximize fitness in changing molecular environments. Synthetic biologists have exploited these capabilities to engineer control on information and energy processing in living cells. While engineered organisms pose important technological and ethical challenges, *de novo* assembly of non-living biomolecular devices could offer promising avenues toward various real-world applications. However, assembling biochemical parts into functional information processing systems has remained challenging due to extensive multidimensional parameter spaces that must be sampled comprehensively in order to identify robust, specification compliant molecular implementations. We introduce a systematic methodology based on automated computational design and microfluidics enabling the programming of synthetic cell-like microreactors embedding biochemical logic circuits, or *protosensors*, to perform accurate biosensing and biocomputing operations *in vitro* according to temporal logic specifications. We show that proof-of-concept protosensors integrating diagnostic algorithms detect specific patterns of biomarkers in human clinical samples. Protosensors may enable novel approaches to medicine and represent a step toward autonomous micromachines capable of precise interfacing of human physiology or other complex biological environments, ecosystems, or industrial bioprocesses.

**Keywords** biochemical programming; computational biochemical circuit design; diagnostics; synthetic biochemical logic circuits; synthetic microreactors
**Subject Categories** Quantitative Biology & Dynamical Systems; Synthetic Biology & Biotechnology
**Mol Syst Biol. (2018) 14: e7845**

## Introduction

From nanoscale biomolecular machines to complex organisms, biological systems have evolved to sense, solve logical problems, and respond to their biochemical environment in optimized ways (Jacob & Monod, 1961). For more than decades, their unique information processing capabilities have fascinated both fundamental and engineering sciences (Feynman, 1960; Conrad, 1985; Bray, 1995). The field of synthetic biology has devoted considerable attention to expanding these biochemical mechanisms into scalable synthetic systems integrating modular biosensing and biocomputing with the hope to advance biotechnologies (Benenson *et al*, 2004; Benenson, 2012; Church *et al*, 2014; Pardee *et al*, 2014; Katz, 2015; Van Roekel *et al*, 2015a; Pardee *et al*, 2016). Indeed, biomolecular machines capable of dynamic probing and decision-making *in situ* could offer new ways to interface biology (Slomovic *et al*, 2015; Courbet *et al*, 2016) as well as unprecedented versatility in analytical and biomedical applications. For instance, synthetic cell-based devices can be designed and employed as programmable bioanalytical tools to detect molecular cues for diagnostic purposes (Courbet *et al*, 2015; Danino *et al*, 2015) or *smart* therapeutics (Ye & Fussenegger, 2014; Perez-Pinera *et al*, 2016; Roybal *et al*, 2016; Xie *et al*, 2016).

While considerable success has been seen in the engineering of control circuits from standardized and composable genetic parts assembled into living cells (Canton *et al*, 2008; Shetty *et al*, 2008; Smolke, 2009), genetically modified organisms have intrinsic limits imposed by the cellular machinery and pose ethical, ecological, and industrial challenges (Chugh, 2013). Instead of repurposing living organisms, biomolecules can be used to build cell-free biochemical circuit-based solutions for information processing (Benenson, 2012). Following success with *in vitro* reconstitution of natural biochemical circuits (Nakajima, 2005), a variety of devices have been designed where biochemical programs are hard-coded within circuits' topology and kinetic parameters ruling the interactions between components, in order to perform useful biomolecular logic: digital/analog circuits (Ashkenasy & Ghadiri, 2004; Niazov *et al*, 2006; Katz & Privman, 2010; Rialle *et al*, 2010; Qian & Winfree, 2011; Orbach *et al*, 2012; Sarpeshkar, 2014; Genot *et al*, 2016), oscillators

1  Sys2diag UMR9005 CNRS/ALCEDIAG, Montpellier, France
2  Department of Endocrinology, Diabetes, Nutrition and INSERM 1411 Clinical Investigation Center, University Hospital of Montpellier, Montpellier Cedex 5, France
3  LRI, Université Paris Sud - UMR CNRS 8623, Orsay Cedex, France
4  EPI Lifeware, INRIA Saclay, Palaiseau, France
5  Institute of Functional Genomics, CNRS UMR 5203, INSERM U1191, University of Montpellier, Montpellier Cedex 5, France
   *Corresponding author. E-mail: acourbet@uw.edu
   †Present address: Department of Biochemistry, University of Washington, Seattle, WA, USA
   ‡Present address: Institute for Protein Design, University of Washington, Seattle, WA, USA

(Semenov *et al*, 2015), switches and memories (Kim *et al*, 2006; Padirac *et al*, 2012), noise filters (Tyson & Novák, 2010), neural networks (Qian *et al*, 2011), or universal Turing machines (Arkin & Ross, 1994) solving hard computational problems (Adleman, 1994; Faulhammer *et al*, 2000; Stojanovic *et al*, 2014). Biochemical signal processing has thus been explored, and metabolic cascades of enzymatic reactions or DNA strand displacement mechanisms have been successfully designed by hand and assembled *in vitro* to yield various useful devices (Sarpeshkar, 2010; Katz, 2012). However, systematically designing arbitrary sequences of logic operations using a variety of biochemical substrates with respect to time-dependent specifications, a strategy we refer to as biochemical programming, has remained challenging. The main reason that has so far prevented the programming of biochemical systems is the exponential growth of the parameter space that consequently cannot be naively sampled to identify robust design implementations.

In the same way as electronic design automation enabled the growth in size and capacity of electronic devices (i.e. Moore's law), automated design frameworks are required to build biochemical control circuits *de novo* (Chandran *et al*, 2011; Chiang *et al*, 2014). Although progress in design automation of synthetic gene circuits has been made (Marchisio & Stelling, 2011; Van Roekel *et al*, 2015b; Delépine *et al*, 2016; Nielsen *et al*, 2016), to date no clear engineering principles or methodologies exist to design cell-free synthetic reaction-based logic systems according to specifications, while using a variety of reactive biochemical species of different nature. Furthermore, in the same way natural cells rely on membrane compartmentalization and localization of metabolons to support complex operations to be performed, microarchitectures are required within which biochemical circuitry can be insulated to allow spatial segregation, parallelization of processes, and multiplexed signal processing (Elani *et al*, 2014). Elegant approaches to synthetic cell-like microreactors containing cascaded circuits of enzymes or nucleic acids, often referred to as *protocells*, have shown recapitulation of complex behavior found in natural cells such as communication, information processing, metabolism, and reproduction (Noireaux & Libchaber, 2004; Caschera & Noireaux, 2014; Sun *et al*, 2015; Adamala *et al*, 2016; Küchler *et al*, 2016; Qiao *et al*, 2016). However, these approaches have so far remained unsuitable for scale-up and for potential use as functional devices, for the reason that the behavior and robustness of manually constructed entities could not be efficiently designed, sufficiently controlled, and maintained (Miller & Gulbis, 2015).

Here, we propose to automate the programming of membrane-insulated synthetic biochemical circuits through computer-aided design and demonstrate that this strategy can be efficiently applied to build biosensing devices that solve bioanalytical problems at the microscale. In our approach, programming a biochemical circuit to exhibit a user-defined dynamic behavior amounts to identifying suitable reactions of kinetically favorable species for processing a signal from input substrates to output product molecules, together with their respective concentrations at which the robustness is maximized. Starting from previously established software suites for modeling and simulating large-scale biochemical systems with low computational requirements (i.e. BIOCHAM, BioNetCAD, and HSIM) (Mazière *et al*, 2004; Maziere *et al*, 2007; Amar *et al*, 2008; Rizk *et al*, 2009, 2011; Peres *et al*, 2010; Rialle *et al*, 2010; Peres *et al*, 2013; Amar & Paulevé, 2015), we scale up the capabilities of computer-aided

design of biochemical logic circuits through the integration of automated implementation relying on a library of parts mined from natural biochemical networks, combined with model checking, sensitivity, and robustness analysis (Koeppl, 2011; Rizk *et al*, 2009, 2011). This enables to automate the search for biochemical circuit solutions to defined logic specifications while providing quantitative *in silico* assessment. Furthermore, we propose to exploit the advantages of digital microfluidic technologies that offer precise control over assembly mechanisms, compartment size and stoichiometry of content, high-throughput generation, and amenability to automation (Miller & Gulbis, 2015). We develop a directed self-assembly microfluidic method that allows us to accurately build picoliter scale cell-like reactors in which biochemical circuits can be insulated within synthetic phospholipidic membranes with respect to *in silico* models.

Using this complete workflow, we show for the first time how to program and assemble *in vitro* discrete synthetic biochemical microreactors that behave according to arbitrary sensing and logic specifications (Fig 1). We coin the term *protosensors*, which we define as minimal cell-like biosensing–biocomputing microreactors that can be biochemically programmed with a wide range of biomolecules or synthetic machinery, into smart and autonomously functioning micromachine (Courbet *et al*, 2017). To our knowledge, our study is the first report of computer-automated design of synthetic cell-like information processing systems.

As a valuable proof-of-concept, we apply our framework to the biodetection of human pathologies. We demonstrate the capabilities of protosensor biochemical programming by implementing a diagnostic algorithm designed to discriminate between all acute metabolic complications of diabetes and achieve differential diagnosis. We further demonstrate the capabilities of this novel diagnostic approach in clinical context and propose that computer-aided biochemical programming of protosensors could provide versatile microscale solutions to complex analytical questions.

## Results

### Operation principles and architecture of protosensors

Our first goal was to identify a universal and robust macromolecular architecture capable of supporting the modular implementation of *in vitro* biosensing/biocomputing processes in the form of synthetic biochemical circuits. This architecture should be capable of (i) stably encapsulating and protecting arbitrary biochemical circuits irrelevant of their biomolecular composition, (ii) discretizing space through the definition of an insulated interior containing the synthetic circuit, and an exterior consisting of the medium to operate in (e.g. a clinical sample), (iii) allowing signal transduction through selective mass transfer of molecular signals (i.e. biomarker inputs), and (iv) supporting accurate construction through thermodynamically favorable self-assembling mechanisms. The protosensor architecture we propose in this study consists of synthetic vesicles made of phospholipid bilayer membranes rendered permeable to small organic molecules through self-incorporation of α-hemolysin transmembrane protein pores.

While nucleic acids have so far been favored due to the advantage of Watson–Crick base pairing-dependent programmability (Padirac *et al*, 2013), we decide to rely on proteins that are versatile

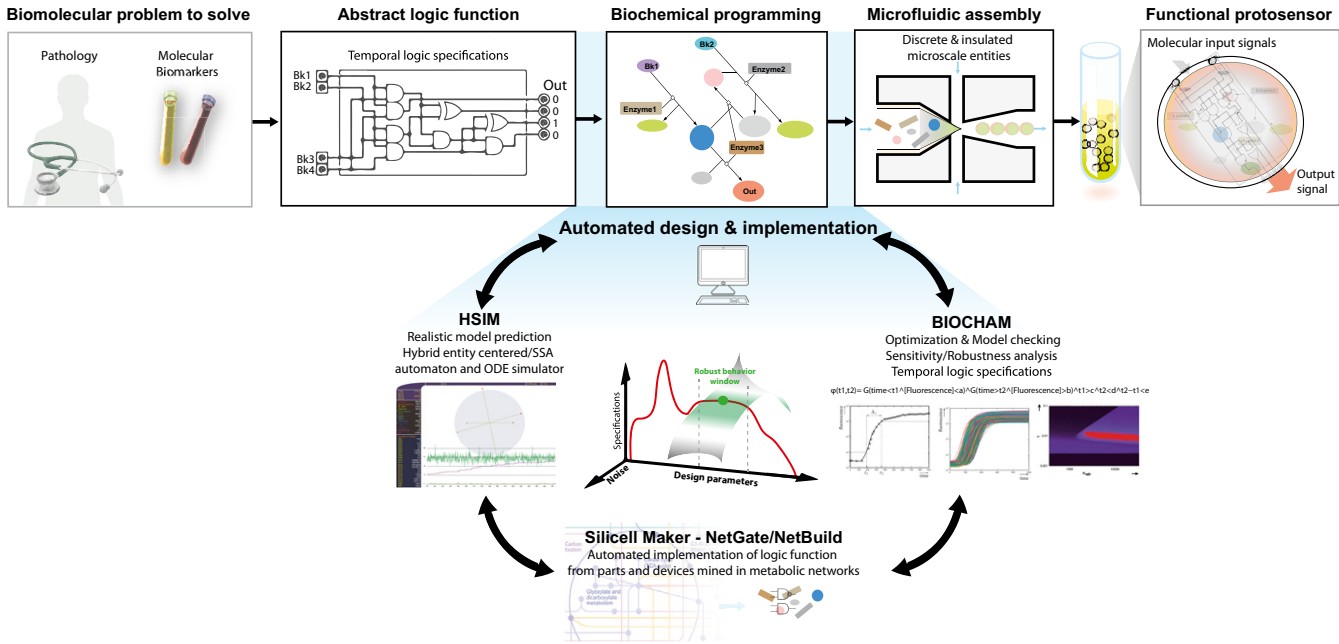

**Figure 1.  General computer-aided design methodology for the programming of synthetic protosensors.**

A specific biotechnological problem, such as assaying for the presence of pathological biomarkers in a clinical sample, can be solved by an appropriate set of combined biosensing/biocomputing operations performed by biochemical species. An abstract Boolean function can be formalized along with a set of kinetic specifications, corresponding to the logic to perform on molecular components *in situ* to solve an analytical and decision problem. Desired Boolean functions can be hard-coded within a biochemical reaction circuit by finding appropriate biomolecular implementations, a process we refer to as biochemical programming. We introduce a systematic methodology based on automated computational design and microfluidics enabling the programming of synthetic cell-like microreactors using *programmed* biochemical logic circuits, or protosensors, to perform accurate and robust biosensing and biocomputing operations *in vitro* according to predefined temporal logic specifications. In order to navigate the multidimensional design space comprehensively, we developed computational tools used to automate the search for synthetic biochemical circuit solutions to formal abstractions. Biochemical circuits are then be experimentally insulated within synthetic membranes to yield autonomous, microscale, discrete protosensors behaving according to specifications.

computational elements offering a wide panel of kinetics and functionality (Bray, 1995). Similarly to natural cells, in this architecture we propose the biochemical work necessary to support signal sensing, processing, and output generation originates from redox reactions. Potential biochemical energy is either stored in encapsulated electrons donors or originates from energy-rich molecular inputs (e.g. glucose). Interestingly, enzyme-gated electron transfer displays useful thermodynamic similarities with current flow in electronics and behaves as elementary biomolecular *transistors* wired through binding kinetics determining logic, signal amplification, and memory (Eyring *et al*, 1981; Mehta *et al*, 2015). Proteins of specific functionality also offer the advantage to be identified through mining of databases of natural biochemical networks, and synthetic biochemical circuits can be easily designed to integrate catalytic activities that depend on specific molecular biomarkers, enabling the coupling of biosensing with decision-making algorithms *in situ*. Here, biochemical information processing under the digital domain to implement decision-making algorithms requires the definition of thresholds for concentration parameters, where a signal at a node of a biochemical circuit is encoded as the continuous valuation or absence of a particular species.

A robust architecture would thus allow us to use a systematic design framework, which is detailed in Fig 1. As a proof-of-concept for the diagnosis of specific human pathologies through the

biodetection of patterns of biomarkers in clinical samples, we chose to implement a clinically useful algorithm enabling to classify acute metabolic complications of diabetes, namely diabetic ketoacidosis, hyperglycemic hyperosmolar state, hypoglycemia, and lactic acidosis, which are known to be associated with a high medical and socioeconomic burden and with important mortality and morbidity (Fig 2A and B).

### Computer-aided biochemical programming of useful algorithms in synthetic biochemical circuits: from *in silico* design to *in vitro* implementation

Programming formal models of biosensing/biocomputing problems amounts to identifying precise biochemical implementations satisfying Boolean logic, molecular input/output, dynamic range, and kinetic specifications within a multidimensional design space. Therefore, a primary key step was to develop a systematic *in silico* framework supporting design automation of synthetic biochemical logic circuits. For this purpose, we developed a computational tool, NetGate, a part of the Silicell Maker software suite, which enables us to mine curated metabolic network databases for biochemical parts, devices, and circuits performing specific Boolean functions, and automates the search for more complex biochemical algorithms (Bouffard *et al*, 2015). In this context, we define mining as the

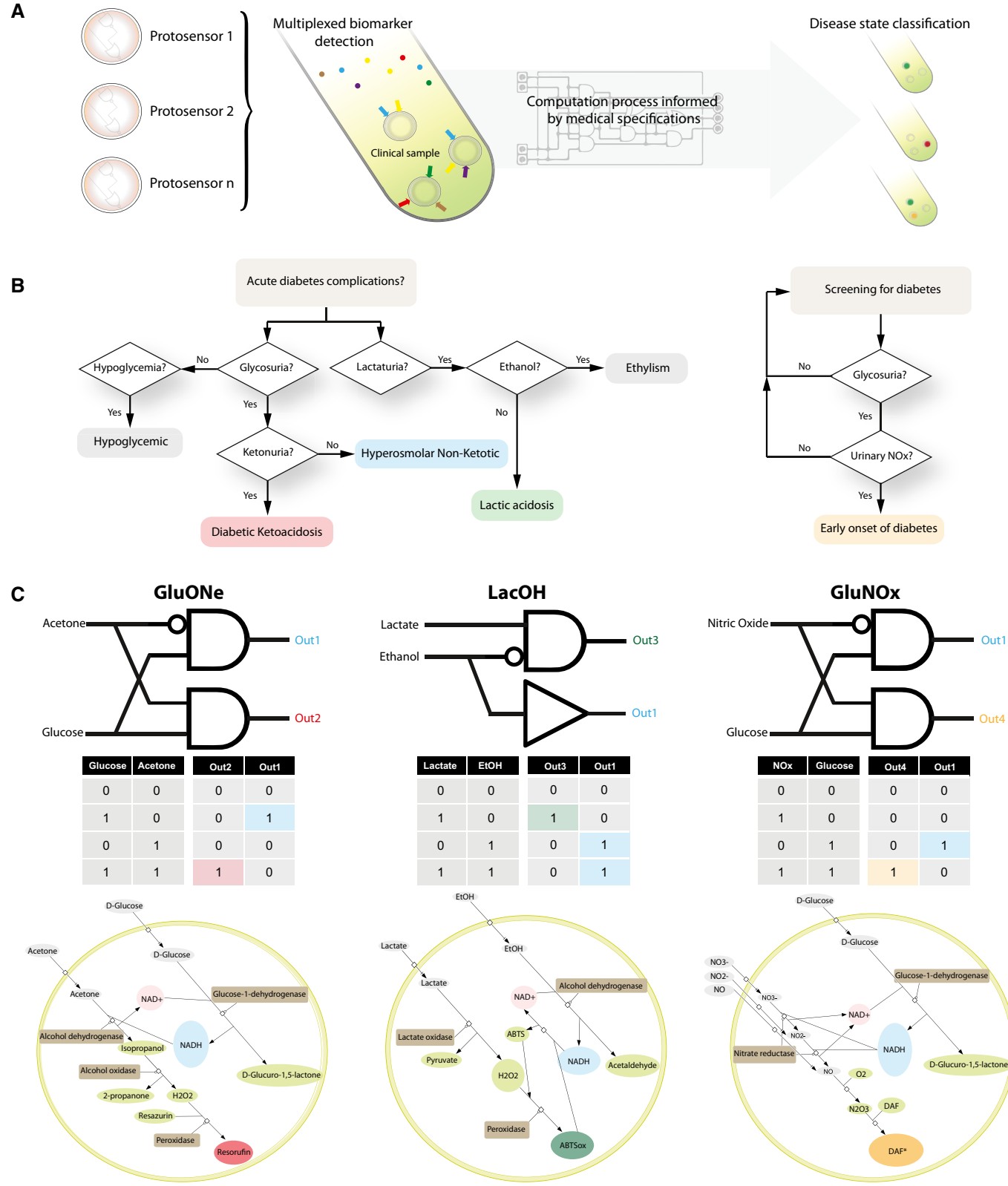

**Figure 2.**

automated implementation of a formal logic function by a set of biochemical reactive species extracted from natural metabolic networks. Since we aim at programming biochemical logic circuits using multiple reactions taking place simultaneously in a microreactor, we reason that mining for molecular species within the same metabolic context *in vivo* would minimize possible failure modes.

**Figure 2. Architecture and operational principle of protosensors for medical diagnosis.**

A   Arising from a clinical need to detect pathologies associated with specific patterns of biomarkers in clinical samples, medical diagnosis can be abstracted to a computational process formalized using Boolean logic *in vitro* and *programmed* into synthetic biochemical circuits. These *de novo* circuits can be programmed and optimized *in silico*, assembled from naturally occurring building blocks, and insulated in synthetic containers *in vitro* to yield diagnostic devices, or protosensors. Protosensors are capable of detecting patterns of specific biomarkers in human clinical samples and integrate these signals in a medical decision algorithm. If a pathological pattern of biomarker is detected, protosensors generate specific colorimetric outputs. Different types of protosensors corresponding to different clinical questions can be used at the same time to enable multiplexed detection of pathological biomarkers, subsequent logic processing, and achieve differential diagnosis of pathologies in clinical samples.

B   Proof-of-concept diagnostic algorithm used in this study and programmed into protosensors circuitry to achieve differential diagnosis of diabetes acute complications and screening for diabetes. Diabetes-associated acute complications, namely diabetic ketoacidosis, hyperglycemia hyperosmolar state, hypoglycemia, and lactic acidosis, are clinical emergencies that represent a major healthcare burden associated with severe mortality, morbidity, and frequent complications. Here, we propose a diagnostic algorithm enabling differential diagnosis of these complications, as well as a proof-of-concept screening assay, from markers present in urines.

C   Logical abstraction and *in silico*-automated implementation of synthetic biochemical circuits for medical diagnosis. Top: formal Boolean description depicted using basic logic gates symbols, and theoretical truth tables for three models recapitulating the medical algorithm, bottom: biochemical circuit solutions found after automated *in silico* search for implementation. SBML models of the synthetic circuits can be found in Appendix.

Briefly, NetGate defines biochemical logic gates by their truth table, the set of molecular species representing input substrates, output products, and enzyme. NetGate takes as inputs a SBML file describing an input metabolic network and a list of truth tables corresponding to Boolean functions that are to be searched in the metabolic network. Additional details about this process can be found in Appendix, while an in-depth description of the algorithm behind NetGate can be found in Bouffard *et al* Second, we developed and refined HSIM (hyperstructure simulator) (Amar *et al*, 2008; Rialle *et al*, 2010; Amar & Paulevé, 2015), a flexible hybrid SSA and entity centered based stochastic and ODEs simulator, which enables fast and accurate model prediction incorporating common biochemical parameters, chemical reactivity (concentrations, Km, Kcat, molecule size, motion, diffusion), and spatial features of microscale structures for realistic physics-based simulation of three-dimensional complex environments. In order to model the selective permeation of small molecules' inputs through the α-hemolysin pores of the protosensors membrane, we implemented in HSIM Fick's equations of diffusion (see Appendix for details). In this study, we use HSIM to perform assessment of kinetically and functionally suitable logic devices circuits and verify the behavior of protosensors. The software environment BIOCHAM (Calzone *et al*, 2006; Soliman, 2012) (Biochemical Abstract Machine) then provides model checking, automated exploration of a multidimensional design space, and optimization of experimental parameters according to temporal logic specifications. Specifically, BIOCHAM supports sensitivity and robustness analysis of the biochemical parameters (e.g. enzyme concentrations) that have to be finely tuned with respect to each other to maximize robustness with respect to specific temporal logic behaviors (Fig 1).

To generate the synthetic biochemical circuits described in this study, we performed an organism agnostic search of all the sets of natural metabolic networks of the BRENDA database with overlapping enzymes, substrates, or products related to the inputs and outputs of the circuits we aimed at designing. SBML files of these networks were then downloaded and merged into a large SBML network (Appendix Fig S1 and Code EV1). This large network was used as input and mined using the program NetGate to identify 775 biochemical logic gates solutions of <2 reactions. The program NetBuild was then used to find unique biochemical implementations satisfying the Boolean logic specifications of protosensors that could execute the particular acute diabetes diagnostic algorithm (Fig 2C, Appendix Figs S1 and S2). The medical algorithm is distributed through three distinct and orthogonal protosensors, each processing two biomarkers as inputs, which were named for convenience GluONe (Glucose and Acetone as inputs, Code EV2), LacOH (Lactate and Ethanol as inputs, Code EV3), and GluNOx (Glucose and Nitric oxides as inputs, Code EV4). The biochemical implementation for these three systems required 6, 5, and 4 different biochemical entities, comprising 4, 3, and 2 different enzymes, respectively. Biomolecular signal processing occurring in these circuits leads to the synthesis of the following measurable output signals molecules: NADH (output 1, 340 nm absorbance), Resorufin (output 2, 571–600 nm fluorescence), ABTS (output 3, 420 nm absorbance), and DAF (output 4, 488–515 nm fluorescence). The Boolean formalism and truth tables corresponding to the medical algorithm, as well as the biochemical implementation, are depicted in Fig 2C (Detailed in Appendix Fig S2), and SBML models of the synthetic circuits can be found in SI.

Feeding HSIM with biochemical knowledge extracted from the BRENDA database, stochastic simulations were then performed to evaluate the behavior of the three circuits (See Appendix Table S1 and S2, Appendix Figs S2 and S3 for more detail). As a first step, we studied models of non-encapsulated synthetic circuits, where initial conditions (i.e. species concentrations) were determined empirically and non-optimized. Predictions of the evolution of these three biochemical circuits after induction with variable concentrations of input biomarkers are represented as computed molecular output signals heat maps (Fig 3A). The relation between computed molecular concentrations and experimental measured signal was calibrated beforehand (See Appendix and Appendix Fig S4). *In silico* models showed that the *de novo* biochemical circuits behaved according to Boolean logic specifications with large signal fold change and near-digital responses. In addition, switching thresholds were found to match useful clinical sensitivity for biomarker inputs (i.e. pathological thresholds: ketones > 17 μM (~10 mg/dl); glucose > 1.39 mM (~25 mg/dl); lactate > 10 μM; EtOH > 17.4 mM (~80 mg/dl)); NOx > 1,000 μM).

Investigation of the experimental behavior of the synthetic circuits prior to encapsulation was then carried out. We proceeded to *in vitro* implementation in the test tube of the previously simulated models with the same initial conditions using recombinant enzymes and synthetic metabolites at room temperature. Multiple behavior mapping experiments consisting in varying input signals were performed and the generated output signals were measured. This allowed us to get a fine understanding of the functioning and

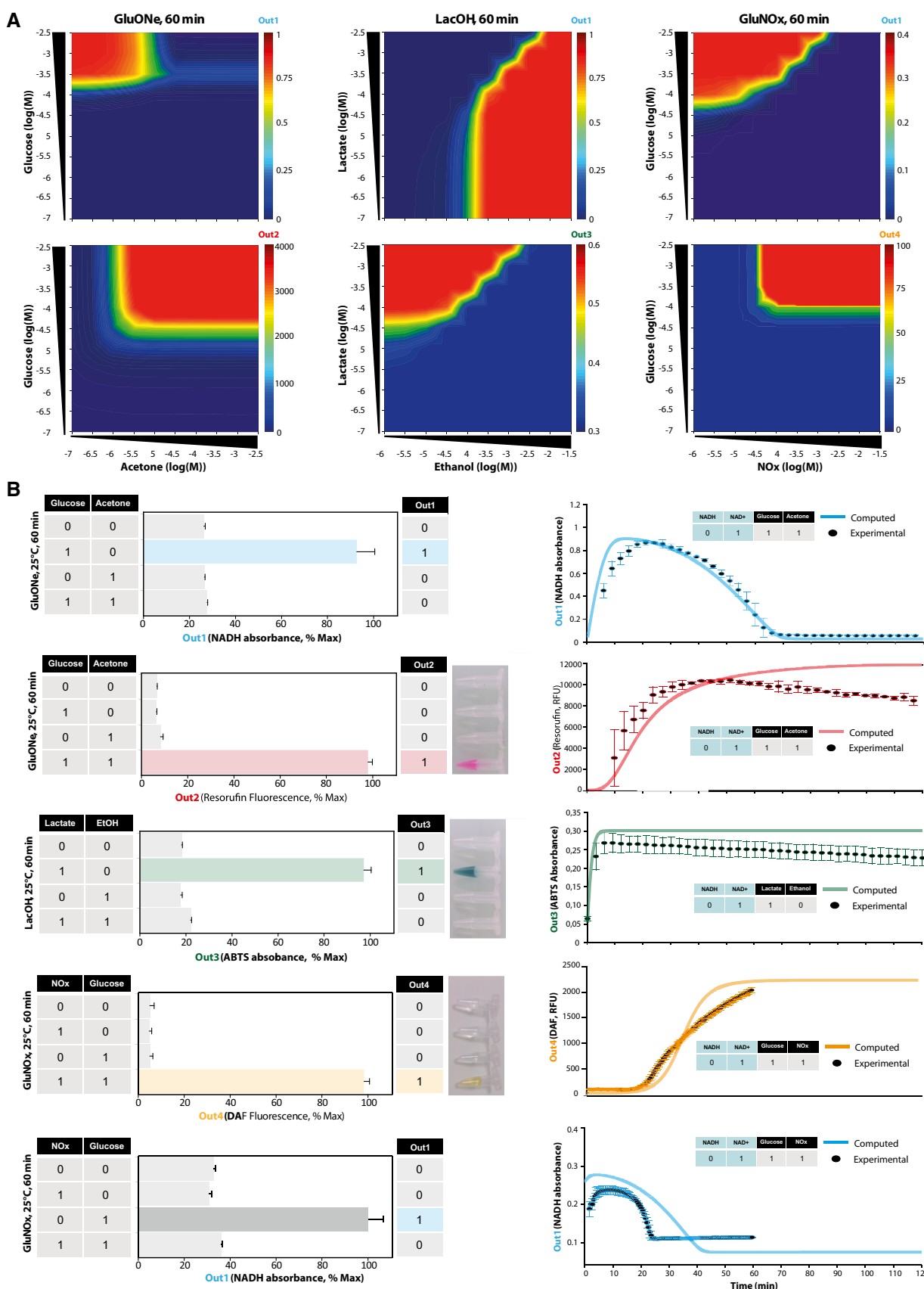

**Figure 3.**

**Figure 3.  *In silico* models and experimental assessment of programmed synthetic biochemical circuits.**

A   Computed heat maps depicting the *in silico* transfer function of non-optimized implementations of synthetic biochemical circuits recapitulating the medical
    algorithm of interest. We used stochastic HSIM simulation to compute the concentration at 60 min, and therefore the absorption (Out1 and Out3) or fluorescence
    (Out2 and Out4) of outputs after induction with varying input concentrations. Concentration parameters were determined empirically, and kinetic parameters were
    extracted from BRENDA database. The mathematical relation between concentration of outputs and related absorption or fluorescence was calibrated beforehand
    (Appendix Fig S4). Heat maps were generated by plotting the mean of five simulations trajectories for each point.
B   *In vitro* biochemical circuits implementations and experimental measurements of truth tables with photograph of tubes at 60 min showing human-readable outputs
    (left), and kinetic behavior compared to HSIM predictions (right). The concentrations of inputs used to induce the circuits were 1 mM acetone, 1 mM glucose, 20 mM
    ethanol, 0.5 mM lactate, 5 mM glucose, and 5 mM NOx. The experiments were carried out at 25°C, and the truth table reflects the values obtained at 60 min. All
    experiments were performed in triplicate wells for each condition and repeated three times on different weeks and different batches, reported is the mean with error
    bars showing SD. From top to bottom, two-sided Student's *t*-test of induced versus highest non-induced condition $P = 1.01286E{-}06$, $P = 1.08059E{-}07$, $P = 1.2022E{-}$
    04, $P = 3.3309E{-}04$, $P = 4.27901E{-}05$.

detailed experimental characterization of logic operations (Fig 3B, Appendix Figs S5 and S6). As expected, kinetic and endpoint measurement showed very good agreement with HSIM predictions, which confirmed the capability of our models to capture relevant biochemical reactivities that condition the behavior of these systems. In addition, the circuits behaved in accordance with Boolean logic specifications with temporal requirements of < 60 min to reach near steady state. Outputs 2, 3, and 4 delivered a human-readable output signal as expected. Considering signal-to-noise ratio (SNR) as a quantitative measure of biocomputing efficacy (Beal, 2015), we found that these synthetic biochemical circuits showed appropriate performance in processing molecular signals according to specified Boolean logic, with calculated SNRs for outputs 1, 2, 3, and 4 of ~20, 34, 14, 26 dB, respectively.

## Protosensor insulation of synthetic biochemical circuits through microfluidic-directed self-assembly

Ensuring picoliter scale control on biochemical parameters is required to achieve programmable protosensors with specified temporal logic properties (Weitz *et al*, 2014). We first investigated the use of previously described phospholipid vesicle fabrication (Pautot *et al*, 2003; Noireaux & Libchaber, 2004; Stachowiak *et al*, 2008; Akbarzadeh *et al*, 2013), which appeared incapable of accommodating precise stoichiometry of various biochemical entities, suffered from either low throughput and poor encapsulation yields. Therefore, we relied on the development of a method that would simultaneously support (i) membrane unilamellarity, (ii) encapsulation efficiency and stoichiometry, (iii) monodispersity, and (iv) increased stability/resistance to osmotic stress.

For this purpose, we developed a custom microfluidic platform and designed PDMS-based microfluidic chips in order to achieve directed self-assembly of a synthetic phospholipid (DPPC) into calibrated, custom-sized membrane bilayers encapsulating low copy number of biochemical species. Briefly, this strategy relied on flow-focusing droplet generation channel geometries that generate water-in-oil-in-water double-emulsion templates (W–O–W: biochemical circuit in PBS—DPPC in oleic acid—aqueous storage buffer with a low concentration of methanol). After double-emulsion templates formation, DPPC phospholipid membranes are precisely *directed* to self-assemble during a controlled solvent extraction process of the oil phase by methanol present in buffer (Fig 4A, experimental details can be found in Appendix and Appendix Figs S7–S10). This microfluidic design also integrates a device known as the staggered herringbone mixer (SHM) (Williams *et al*, 2008) to enable efficient passive and chaotic mixing of multiple *upstream* channels under

Stokes flow regime. We reasoned that laminar concentration gradients could prevent critical mixing of biochemical parts, precise stoichiometry, and efficient encapsulation. We hypothesized that synthetic biochemical circuits immediately homogenized before assembly could standardize the encapsulation mechanism and reduce its dependency on the nature of insulated materials. Moreover, this design allowed for fine-tuning on stoichiometry via control on the input flow rates, which proved practical to test different parameters for straightforward prototyping of protosensors.

We used an ultrafast camera to achieve real-time monitoring and visually inspect the fabrication process, which allowed to estimate around ~1,500 Hz the mean frequency of protosensor generation at these flow rates (Appendix Fig S10 and Movie EV1). A strong dependence of protosensors generation yields on flow rates was found, which we kept at 1/0.4/0.4 μl/min (storage buffer/DPPC in oil/biochemical circuit in PBS, respectively) to achieve best assembly efficiency. We then analyzed the size dispersion of protosensors using light transmission, confocal, and environmental scanning electron microscopy (Fig 4B, Appendix Figs S8 and S9). Monodispersed protosensors with average size parameter of 10 μm and an apparent inverse Gaussian distribution were observed. Interestingly, biochemical circuit insulation did not appear to influence the size distribution of protosensors, which supports the decoupling of the insulation process from the complexity of the biochemical content. Moreover, no evolution of sizes was recorded after 3 months, which demonstrated the absence of fusion events between protosensors. In order to assess the capability of protosensors to encapsulate protein species without leakage, which is a prerequisite to achieve rational design of biochemical information processing, we assayed encapsulation stability using confocal microscopy. To this end, an irrelevant protein bearing a fluorescent label was encapsulated within protosensors, and the evolution of internal fluorescence was monitored over the course of 3 months. The internal fluorescence was found to remain stable, which demonstrated no measurable protein leakage through the protosensor membrane hemolysin pores in our storage conditions. In addition, using phospholipid bilayer specific dye, $DiIC_{18}$, which undergoes drastic increase in fluorescence quantum yield when specifically incorporated into bilayers (Gullapalli *et al*, 2008), the complete extraction of oleic acid from the double emulsion and a well-structured arrangement of the bilayer could be visualized (Fig 4C). We next sought to assess the encapsulation of biological enzymatic parts inside protosensors. Two relevant enzymes were insulated within protosensors: alcohol oxidase and glucose-1-dehydrogenase, and UPLC-ESI mass spectrometry analysis was performed on the protosensors. We found that we could retrieve the molecular signatures of the enzymes in the interior of

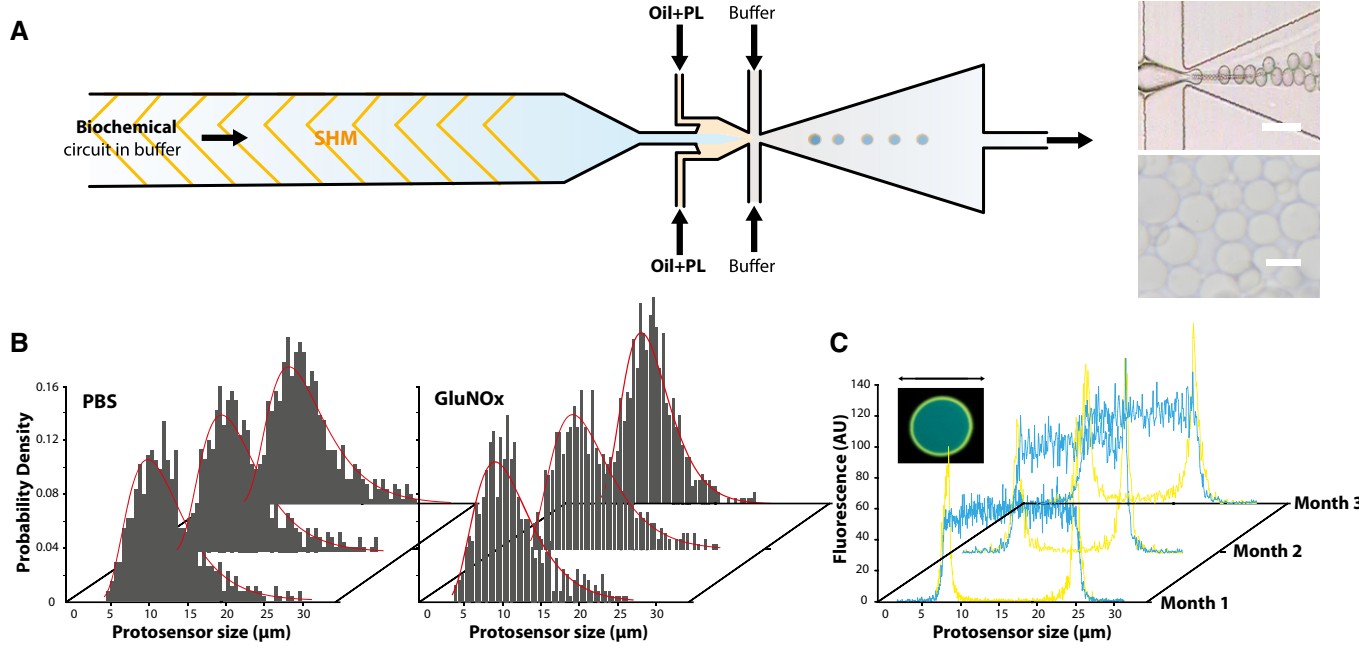

**Figure 4. Experimental construction of protosensors using microfluidics.**

A   Left: Double-emulsion templating microfluidic device architecture and operation. This method relies on the generation of W/O/W double emulsions. Buffer: 10% v/v methanol, 15% w/v glycerol, 3% w/v pluronic F68 in PBS (1 μl/min) Oil+PL: DPPC dissolved in oleic acid (0.4 μl/min) Biochemical circuit buffer: enzymes and metabolites in PBS (0.4 μl/min). Right: microscopic validation of protosensor generation *on chip* (top, scale bar = 20 μm), bright field optical validation of protosensors isolated for subsequent analysis (bottom, scale bar = 10 μm).

B   Size dispersion and stability of protosensors generated with the microfluidic method, after encapsulating PBS (left) and GluNOx network (right).

C   Months-long stability of encapsulation of protein components within the synthetic membrane. AlexaFluor-488-labeled IgG where encapsulated in the DPPC membranes using the microfluidic setup and fluorescence vesicles was then monitored by confocal microscopy for 3 months. At each time point, individual vesicles were isolated from the same solution and observed under confocal microscopy, and representative fluorescence properties are here depicted. Yellow: membrane fluorescence as labeled with phospholipid dye DiIC$_{18}$. Blue: AlexaFluor-488-labeled IgG fluorescence.

protosensors, as compared with positive controls (Appendix Fig S11). Taken together, these findings show that this setup proved capable of generating stable, modular protosensors with high efficiency, and user-defined finely tunable content.

### Biochemical programming of diagnostic protosensors: from *in silico* optimization to digital signal processing, multiplexing logic, and analytical evaluation

We then proceeded to *in silico* optimization of the three medical protosensors implementations before *in vitro* assembly and experimental assessment. More specifically, the initial state concentration parameters of the species constituting the insulated synthetic circuit needed to be optimized to account for temporal logic in accordance with the medical algorithm of interest. Additionally, selective membrane permeability parameters were incorporated in our models, describing passive diffusion of molecular inputs through hemolysin membrane pores (see Appendix for details). The concentration of hemolysin is expected to play a major role in the protosensors response, since it impacts the diffusion kinetics and therefore the kinetics of the whole protosensors responses. In this study, we fixed the concentration of hemolysin to an arbitrary value for which biochemical circuits are optimized by computing the concentration parameters that satisfy chosen temporal logic specifications. We defined temporal logic specifications satisfying clinical

requirements, that is obtaining biosensing sensitivities at pathological thresholds, achieve specified signal processing operations and obtain a measurable output signal in < 10 min for the three systems (See Appendix for details). Feeding synthetic circuits with non-optimized concentration parameters in BIOCHAM, sensitivity analysis was first performed on the models to determine which concentration parameters had the most important influence on protosensors behavior. Central to this approach is the notion of satisfaction degree of temporal logic formulae. This continuous measure of LTL (R) Formulae can be computed to serve as a fitness function in order to find biochemical kinetic parameter values satisfying a set of biological properties formalized in temporal logic (Rizk *et al*, 2008, 2009), thereby evaluating numerically the adequateness of a model relative to temporal logic specifications. The satisfaction degree is normalized such that it ranges between 0 and 1, with a satisfaction degree equal to 1 when the property is true and tending toward 0 when the system is far from satisfying the expected property.

For each protosensor models, validity domains were computed to extract concentration thresholds (N and R) at steady state (T) satisfying temporal logic specifications. We first performed sensitivity analysis on concentration parameters with a specified logic formula corresponding to the desired systems specifications. This permitted us to identify the two most sensitive concentration parameters of the systems. Comprehensive 2D sensitivity landscape maps of configurations satisfying specifications relative to these two dependencies

were then computed in order to visualize the available biochemical design space (Fig 5A, Appendix Fig S12). For each system, precise parameter spaces boundaries satisfying temporal logic could thus be identified. GluONe protosensors (Code EV5) operation appeared mostly sensitive to G1DH and ADH enzymes concentration. Interestingly, the behavior of LacOH (Code EV6) and GluNOx (Code EV7) appeared more sensitive to the initial concentrations of the metabolite $NAD^+$ than other enzymes. For all three systems, the $NAD^+$/NADH redox ratio can be seen as a biochemical *current* connecting the two molecular inputs signals and thus has to be finely tuned to match input thresholds and enzyme levels. Within the computed design space, initial state concentrations could be then rationally chosen using BIOCHAM automated parameter search (CMAES Method), in order to optimize robustness of operation while satisfying temporal logic specifications according to each model. In addition, arbitrary implementations outside of the *in silico*-validated specification landscape yielded non-functional systems as verified *in vitro* (Fig 5A, Appendix Fig S12). In order to further verify protosensors behavior *in silico*, the transfer function was mapped using stochastic HSIM simulations. As previously, we generated heat maps of computed output signals after induction with various concentrations of biomarker inputs (Fig 5A). Boolean logic functions were found to be respected with appropriate theoretical response fold change, as well as near-digital response profiles. Theoretical switching thresholds for these models were also found to match useful clinical sensitivity for biomarker inputs. We then sought to investigate the behavior of protosensors experimentally in detail and proceeded to microfluidic *in vitro* assembly of GluONe, LacOH and GluNOx protosensors using computer-optimized parameters as previously defined. We performed multiple experiments consisting in mapping the experimental truth tables of the protosensors. Output responses at the population level were measured while varying input conditions (i.e. presence/absence of pathological concentrations of input biomarkers). This allowed us to get a fine

understanding of the functioning and detailed experimental characterization of logic operations (Fig 5B, Appendix Figs S13 and S14). Clear digital-like behaviors with important fold changes were obtained that showed exact accordance to Boolean logic specifications with temporal requirements of < 60 min. The calculated signal-to-noise ratio showed very good performance SNRs for outputs 1, 2, 3 and 4 of ~8, 35, 5, and 11 dB respectively.

Not only capable of recapitulating the programmed Boolean operations, we reasoned that a validation of our model predictive power would be to achieve the same transfer functions *in vitro* as previously predicted by simulations. In a second set of experiments, the three protosensor systems were incubated with increasing concentrations of respective input biomarkers, and their individual output signal response was measured using flow cytometry. We hypothesized that this technique would give precise single *(proto)* cell level measurements, thus lowering sample noise effects (Fig 5C). When comparing these data to HSIM model simulations, experimentally measured switching thresholds were found to show very good agreement with predictions, along with near-digital responses. In order to further validate the spatial and analytical digitization of output signals, confocal microscopy was used to quantitatively measure and precisely visualize output signals generation within single induced protosensors (Fig 5D). Bright images with high SNRs and important response fold changes were obtained, strongly corroborating previous flow cytometry data. Molecular output signals appeared well localized in the interior of protosensors, although possible leakage was not quantified.

Although satisfying for analytical applications, greater background noise was found compared to non-encapsulated synthetic biochemical circuits. We hypothesized that this was due to autofluorescent species such as surfactant and phospholipids, along with probable scattering and absorbance phenomena emerging for spherical protosensor structures in solution. The rationale behind encapsulation was to achieve greater analytical robustness and obtain

**Figure 5.** ***In silico* optimization and experimental validation of analytical properties of protosensors performing robust multiplexed biosensing and logic.**

A  Mapping satisfaction degree landscape for GluOne (top), LacOH (middle), and GluNOx (bottom) protosensors. Satisfaction degrees of temporal logic formulas at 5 min were computed while varying the two most sensitive parameters of respective models (e.g. ADH and G1DH concentration for GluONe), for each combination of inputs. Temporal specifications of output concentration thresholds at steady state along with validity domains are depicted below each map. Optimized concentration parameters were then computed using CMAES method implemented in BIOCHAM. We verified experimentally the kinetic response of protosensors implemented with specific concentration of enzymes corresponding to the colored squares on the map. We then computed the heat maps depicting output signals at 60 min using HSIM models simulations, for the three different concentration optimized systems after induction with increasing biomarker concentration. Heat maps were generated by plotting the mean of 5 simulations trajectories for each point.

B  Experimental truth tables of protosensors operating in human urines. The concentration of inputs used for induction corresponded to pathological threshold defined as 17 μM acetone, 1.39 mM glucose, 10 μM lactate, 17.4 μM ethanol, and 1 mM NOx. The experiments were carried out at 25°C, and the truth table reflects the values obtained at 60 min. From top to bottom, two-sided Student's *t*-test of induced versus highest non-induced condition $P = 3.3627$ E-06, $P = 1.1701$ E-04, $P = 4.2909$ E-06, $P = 1.3397$ E-05

C  Experimental validation of computer prediction at the single (proto)cell level using flow cytometry. We measured the main fluorescence output of each different protosensor logic systems 60 min after induction by the respective biomarkers at discrete inducer concentrations as indicated for each corresponding contour plot. For LacOH, in order to get a fluorescent output signal measurable with a flow cytometer, we exchanged ABTS output with Resorufin, which both undergo analogous reduction process. Flow cytometry data collection was carried on 10,000 individual events.

D  Confocal microscopy validation of "ON" output signals responses at 60 min after induction with respective biomarkers, corresponding to concentrations indicated in the left and right dashed box of (c). From top to bottom: GluONe (Out2), LacOH (Out3), and GluNOx (Out4). The phospholipid bilayer was stained in yellow using the dye DiC$_{18}$. For LacOH, in order to get a fluorescent output signal measurable in confocal microscopy we exchanged ABTS with Resorufin, which undergo an analogous reduction process.

E  Multiplexing Logic: Example of comparison between expected (valid analytical response according to molecular inputs present in the sample), batch mode analysis (non-encapsulated synthetic circuits), and protosensors analysis (encapsulated synthetic circuits). PBS media were spiked with multiple biomarkers (in this case, acetone, ethanol and nitric oxides or glucose, acetone, and lactate), and output signals were measured after 60 min.

Data information: All experiments were performed in triplicate wells for each condition and repeated three times on different weeks and different batches; all data points are shown as mean with error bar showing SD.

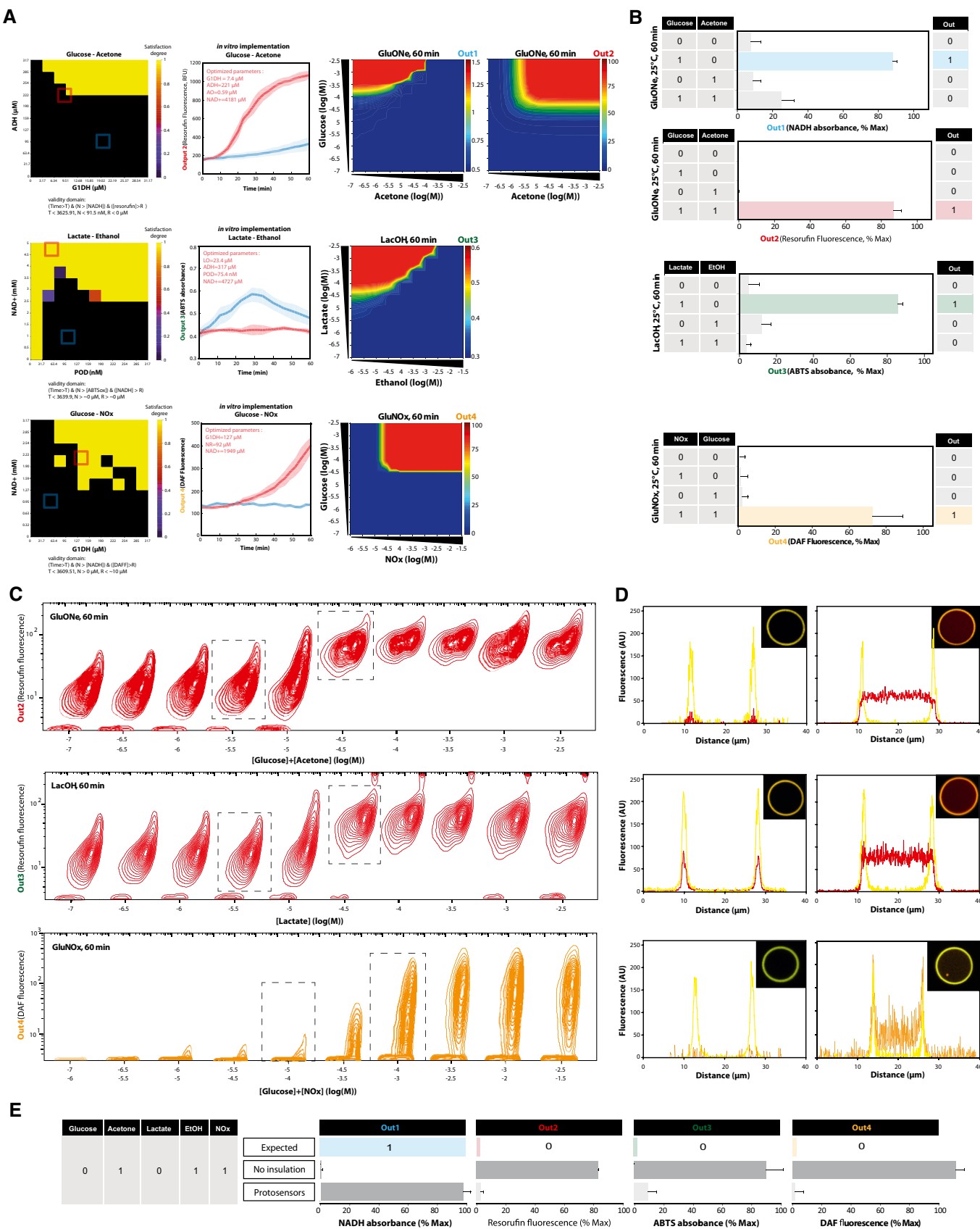

**Figure 5**

insulation to achieve programmable Boolean logic. In other words, multiple types of protosensors in solution should facilitate design, and be able to operate independently in a standardized way without interacting with each other in order to achieve multiplexed analysis of the molecular environment. To verify this assertion, we addressed multiplexing logic capabilities experimentally. For this purpose, different experiments were set up where we measured output signals in media spiked with multiple biomarkers, by mixing all three synthetic biochemical circuits at the same time in either *batch* mode analysis (i.e. non-encapsulated synthetic circuits) or protosensors mode analysis (i.e. membrane-insulated synthetic circuits). For this experiment, arbitrary combinations of biomarkers were chosen (Fig 5E). The measured output signals obtained were compared to the expected output corresponding to the programmed specifications. Mixing non-insulated synthetic biochemical circuits were found incapable of achieving correct signal processing tasks, most likely due to molecular interactions between the circuits' components. On the other hand, mixing the three types of protosensors did not affect biosensing and signal processing capabilities, which were capable of performing Boolean logic and output signal generation.

### Assaying pathological clinical samples: protosensors mediated diagnosis of diabetes

After successful validation of protosensors analytical capabilities in spiked samples, we then sought to perform real-world assessment for disease detection in a clinical setup. In a clinical perspective, we first sought to address the potential effects of clinical samples, such as urine media, on protosensor architecture and operability (Fig 6A). We reasoned that measuring protosensor fluorescence signal along with forward-scattered light (FSC) using flow cytometry would give insights on protosensor stability, as FSC is correlated with vesicular size and internal fluorescence with membrane

integrity. Induction and prolonged incubation in urine was not found to impact stability, structure, or operability of protosensors. In addition, no difference between operation in standard PBS buffer and urine media was apparent (Fig 6A and Appendix Fig S15). Taken together, these data demonstrate that protosensors support the implementation of programmed biosensing and biomolecular logic gated operations irrelevant of *in situ* context, with robust and predictable behavior.

GluONe protosensors were then evaluated as a proof-of-concept to detect endogenous levels of pathological diabetes-associated biomarkers in clinical samples from patients. Although we did not benefit from a large sample library of diabetes-related metabolic complication to test the complete implemented diagnostic algorithm, previously collected urine samples from treatment-naive diabetic patients were available. We reasoned that assaying pathological glycosuria and absence of ketonuria in these urine samples would constitute a testbed diagnostic evaluation. We proceeded to incubation of GluONe protosensors with either diabetic urine samples or non-diabetic control urines and as previously described measured output signal responses (Fig 6B). Glycosuria analysis using the clinical point-of-care gold standard was also concomitantly performed (i.e. urinary dipsticks). We found high correlation between output signals from protosensors and visual examination of dipsticks. Moreover, receiver operating characteristic (ROC) analysis of the data showed that the assay reliably detected glycosuria in samples from diabetic patients, with a near ~100% sensitivity and specificity, and an area under curve of ~0.9981, which defines GluONe protosensors as high performance diagnostic test holding comparison to gold standard. Taken together, these data demonstrate that protosensors can discriminate between normal and diabetic patients with excellent diagnostic accuracy. Therefore, we conclude that rational biomolecular programming of protosensors may be used to generate clinical-grade assays to detect endogenous biomarkers of disease in patient samples.

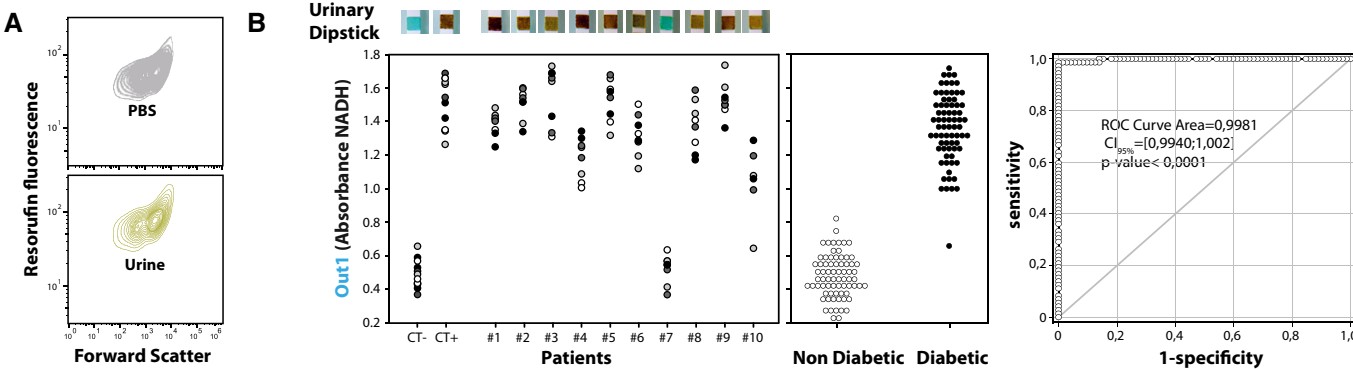

**Figure 6. Protosensor-mediated detection of pathological glycosuria in patient clinical samples.**

A  Flow cytometry evaluation of protosensors structural robustness in urines. GluONe protosensors were induced and incubated for 2 h at 25°C in human urine and analyzed using flow cytometry while recording Resorufin fluorescence and forward-scattered light.

B  Left: Dot histogram of data used for statistical analysis. Dots of the same color (white, gray, black) correspond to one repetition of the experiment, which was performed in triplicates in different patient samples. Above each data point is depicted a photograph of a urinary dipstick measurement performed on the sample. We then plotted a dot histogram of data comparing the non-diabetic to diabetic set. Right: Receiver operating characteristic (ROC) analysis curve depicting statistic sensitivity versus (1-specificity). A set of 72 measurement performed in non-pathological urine were compared to 72 measurements performed in urine from diabetic patients. The data were processed using the software SigmaPlot (Systat Software Inc.) with a computed $P < 0.0001$ using a Mann–Whitney–Wilcoxon test. We used GluONe protosensors for this assay and measured Out1 signal (NADH absorbance) after 60 min at room temperature.

# Discussion

A vast landscape of problems in biology and medicine has remained unsolved due to our inability to rationally engineer synthetic information processing biomolecular systems. The last decades witnessed a growing interest for the study of synthetic biological control circuits, largely inspired by application pervasiveness of portable, autonomous, and programmable biosensors and biocomputer devices. *In vivo* approaches have traditionally been favored due to our increasing ability to re-engineer molecular machinery of living organisms using a variety of new biotechnologies. However, developing novel frameworks to rationally build *de novo* at the system scale using biomolecular components could open up the way for tremendous bioengineering and biomedical applications.

Our study builds upon previous attempts at assembling biochemical circuits *de novo* to perform useful biomolecular logic operations. Here, we brought rational design of *in vitro* synthetic biochemical circuit closer to real-world applications by addressing some of the previous technological limitations, namely design, programmability, and scalability. We developed a systematic computational approach combining automated exploration of the biochemical design space according to time-dependent quantitative and qualitative specifications, to automated robustness assessment and optimization of initial concentration parameters. To our knowledge, our *in silico* framework is the first step ever made toward task-oriented programming of synthetic biochemical circuits while providing quantitative evaluation of functionality and analytical properties to maximize reliability in operation. We provide evidence to support the role of model checking as a key enabling methodology that allows for navigation of the design space and biochemical implementation of logic and specifications, which is otherwise prone to failure.

Although nucleic acids had been mostly used for this purpose for the ease of Watson–Crick programmability (Chen *et al*, 2013), here we report for the first time a systematic method to extend biochemical programming to protein catalysts and metabolites. Additionally, we developed an accurate microfluidic-directed self-assembly methodology capable of supporting the accurate, picoliter scale implementation, and insulation of biochemical circuits within synthetic membrane boundaries. Using a combination of these methods, we showed that we could synthesize *in vitro* microscale devices, protosensors, combining biosensing and biocomputing abilities, capable of sensing their molecular environment along with biomolecular signal processing and decision-making in user programmable ways. We demonstrated that membrane insulation provides a robust architecture to decouple the biochemical *software* from the complex medium it operates in. In comparison with previous attempts, this approach could increase the scale and the complexity of the circuitry that can be programmed by layering modules and preventing deleterious molecular electron transfer *short-circuits* (Agapakis & Silver, 2010). Enzymatic activities susceptible to perturbations could thus be confined, programmed in circuits, standardized, and assembled in discrete sensing and computing units, which operation and failure modes are easier to model *in silico*. Larger scale circuits could in the future be built by programming different protosensors coordinating a common behavior at the population level.

Nevertheless, computer-aided programming of protosensors may not be as straightforward for insulated circuits of larger size. Indeed,

predictive computer models may still fail due to oversimplification of molecular interactions and incomplete knowledge of reactivity and physicochemical properties of species generating unpredictability arising at the system level. Most biological parts available for the bottom-up engineering of synthetic biochemical circuits remain insufficiently characterized in increasing context complexity. Moreover, a lack of orthogonal parts may still limit the scaling up of *in vitro* synthetic circuits. For instance, some input biomarkers may in certain case lack biochemical sensors interfacing with signal processing modules. In order to overcome these limits, our current efforts focus on eliminating the remaining steps of human intervention and further automate classification, standardization, and robustness assessment of biological parts in context. This initiative is underway through automated refinement of databases, which stores parts, abstract modules such as sensors, switches, or Boolean logic gates extracted from biological networks (Bouffard *et al*, 2015), which we hope will lead to constant improvement in bottom-up design capabilities. Efforts could be directed toward augmenting the models' depth of description from Brownian and Michaelian kinetics to increasing molecular mechanistic (Mazière *et al*, 2004; Maziere *et al*, 2007). Additionally, here we applied Boolean logic familiar to electrical engineering to the biochemical substrate in order to achieve programmable biological information processing. Nonetheless, other modes of operation closer to natural signal processing have been described, such as analog computation (Sarpeshkar, 2014). Future efforts focusing on implementing hybrid analog–digital strategies in biochemical circuits would be greatly valuable.

The current architecture relying on phospholipid bilayer membranes may impose some intrinsic limits. Although we assessed stability and impermeability of the membrane, which is a prerequisite to achieve biochemical signal processing by preventing leaking of components in and out of the protosensors, we did not yet address the biochemical stability of the enzymes. However, many studies have explored this experimentally and showed that encapsulation of enzymes enhance their protection against denaturation and proteases (Küchler *et al*, 2016).

We also have yet to explore the consequences of osmotic stress in challenging environments that could compromise membrane integrity and find ways to achieve fine-tuning of selective signal transduction through the membrane. Although phospholipidic membrane could prove useful in some specific applications, we envision to confine their use as a proof-of-concept model of microreactor confinement. We envision that more robust insulation mechanisms may be needed to support applications necessitating osmotic pressure resistance. We envision that harnessing nanoscale self-assembly mechanisms, for instance using synthetic protein nanomaterials such as nanocages (King *et al*, 2012, 2014; King & Lai, 2013; Bale *et al*, 2016) or other multidimensional architectures (King & Lai, 2013; Yeates *et al*, 2016), could provide valuable alternatives as modular compartments for biosensing–biocomputing architectures. Similarly, instead of exploiting natural catalytic activities, synthesis of orthogonal biochemical parts could be used to design future systems. For instance, computational protein design can be used to design tailored enzymes with novel non-natural catalytic functions (Siegel *et al*, 2010, 2015; Kiss *et al*, 2013; Heinisch *et al*, 2015; Huang *et al*, 2015, 2016).

In this study, we demonstrated that protosensors are promising tools to perform *in vitro* diagnostics integrating medically relevant

algorithmic processes. Protosensors offer multiplexed detection and sophisticated analytical capabilities with sharp near-digital response profiles coupled to expert decision-making. As a prototype, we showed that this technology could be successfully applied to solve clinical problems such as the diagnosis of diabetes-related complications. Not only smartening classical *in vitro* diagnostics by integrating expert decision-making, protosensors could be used for cheap, portable, and multiplexed screening of complex pathophysiology at the point of care, which remains a limiting challenge of current diagnostics. In particular, we envision that the diagnostic capabilities, programmability, and versatility of these devices could greatly benefit precision medicine agendas or the management of rare or rapidly emerging pathologies. Protosensors could support delocalized uses and modes of detection for dynamic biomarker signatures, and thus bear potential for the management of complex syndromes. Furthermore, we suspect this approach to biosensing, which relies on autonomous and programmable entities, to be of interest for novel kinds of local measurements and bioactuation since protosensors can be addressed to specific biological structures or cells *in vivo* through external receptors. These systems could be further engineered into *sense-act* micromachines, for instance conditionally generating actuation signals or therapeutic responses *in situ*, as well as interfacing or integrated in living systems. Communication capabilities could as well be integrated in design as previously proposed (Adamala *et al*, 2016), potentially allowing protosensors to achieve complex cooperative behaviors. Last, protosensors may be amenable to spatial patterning and ultra-high-throughput applications through high-density *chip* immobilization.

This study paves the way for the systematic programming and extended use of integrated biochemical circuits in protocellular structures. We adapted for the first time computational methods of engineering sciences such as model checking to bottom-up synthetic biology, which we envision could drive interest to the field. This approach could offer avenues to test and validate biomolecular circuitry, and thus provide new insights on the fundamental principles governing biological information processing. These automated methods can be used to evaluate how a biomolecular system satisfies a set of properties expressed in temporal logic, and as such could be extended to a wide range of applications in synthetic biology. For instance, we envision that our tools could prove useful in metabolic engineering for *de novo* pathway construction and flux optimization (Dudley *et al*, 2015). In a long-term vision, we envision the establishment of bottom-up biochemical circuit programming as a universal framework to produce a wide array of tools for biomedical research and the industry, from probing and interfacing biological structures, cellular reprogramming, ultra-low-power biomolecular computing, and the rational integration of biological parts toward the synthesis of minimal autopoietic systems.

# Materials and Methods

### Study design

To evaluate the robustness of our system and its functionality in clinical samples, we used urine pools from healthy individuals as well as urine samples from diabetic patients. Regarding collection of

clinical samples, non-pathological (control) and glycosuric (diabetic) urine samples were obtained from the Department of Endocrinology of the Lapeyronie Hospital, Montpellier, France, under the supervision of E. Renard. Individual informed consents were obtained from the patients and control individuals and the experiments conformed to the principles set out in the WMA Declaration of Helsinki and the Department of Health and Human Services Belmont Report. Glycosuric urine samples were collected from 10 newly discovered, non-stabilized diabetic patients. Urine samples were stored at −80°C before use.

### Enzymes and biochemistry

All enzymes, phospholipids, and chemicals were purchased from Sigma-Aldrich, stored at −80°C, and used in PBS at pH 7.4 at 25°C. Concentrations used, initial conditions, experimental steps, and detailed information can be found in Appendix.

### *In silico* design and simulation

Computer-automated implementation of biochemical logic circuits was performed using the Silicell Maker software suite with NetBuild and NetGate programs (https://silicellmaker.lri.fr/, Patrick Amar), and stochastic simulations (SSA) were performed using HSIM (https://www.lri.fr/~pa/Hsim/, Patrick Amar). Recent improvements that we brought to HSIM for the purpose of this study consist of multisubstrate enzymatic mechanisms implementation (ordered sequential bi–bi mechanism, Ping Pong bi–bi mechanism, and random sequential bi–bi mechanism), automated translation of probability from HSIM stochastic simulator to mass action rates for BIOCHAM ODE solver, the implementation of physical models for protosensor membrane permeability, as well as SBML support (more information on these improvements can be found in Appendix). Simulations were run for 60 min, with protosensors diameters of 10 μm, and computed output values were calculated using calibration curves giving relation between concentration and fluorescence or absorbance values. Permeability coefficient and other kinetic parameters were obtained from previous literature and the BRENDA database. Sensitivity analysis and parameter search were performed using BIOCHAM (https://lifeware.inria.fr/biocham/). Step-by-step detailed information, computer code, and BIOCHAM executable notebooks showing the process of biochemically programming a protosensor can be found in Appendix and can be executed on the online server (http://lifeware.inria.fr/biocham/online/). SBML models (Code EV1–EV4), and corresponding Biocham code (Code EV5–EV7) of the synthetic circuits described in this can be found in SI. Latest version of software executable can be found on the link featured in this section.

### Protosensors microfluidic generation

PDMS microfluidic chips were designed and prototyped using AutoCAD software, and fabrication was carried out by the Stanford University microfluidic foundry. The microfluidic chips were connected with PTFE tubing to neMESYS V2 syringe pumps (Cetoni GmbH, Germany). Microfluidic processes were imaged using a Leica DMIL microscope mounted with a Canon 750D or a Phantom V7.3 camera. Detailed information can be found in Appendix.

## Spectrometric assays, flow cytometry, and microscopy

To test the operability of the systems, synthetic biochemical networks and protosensors were inoculated in 100 µl total volume of PBS with or without input biomarkers, or urine from patients diluted at a ratio of 1:2 in PBS for a total volume of 100 µl in a 96-well plate. Kinetic measurements or endpoint measurements after 1 h of incubation were performed at 25C with gentle 200 cpm double orbital shaking; fluorescence and absorbance were read using a synergy H1 plate reader (more details can be found in the Appendix). We concomitantly tested these urines from non-stabilized diabetic patients using the Siemens Multistix 8 SG reagent strip according to the manufacturer's protocol. Flow cytometry experiments were performed on a Guava EasyCyte benchtop flow cytometer (Merck) equipped with a 488-nm laser and analyzed using FlowJo vX software. Confocal microscopy assays were performed on a Leica SP8-UV equipped with 63× oil lens and 355-, 488-, and 561-nm lasers. All experiments were performed in triplicate wells for each condition and repeated three times on different weeks and different batches.

## Data analysis and statistics

Experimental values are reported as means ± SD. For all bar plots, reported $P$-values were computed used a two-sided unpaired $t$-test. All experiments were performed in triplicates. Data, statistics, graphs, and tables were processed and generated using MATLAB (MathWorks) and SigmaPlot (Systat Software Inc.). Flow cytometry data collection was carried on 10,000 individual events. For receiver operating characteristic analysis, a set of 72 measurements performed in non-pathological urine were compared to 72 measurements performed in urine containing 1% glucose.

**Expanded View** for this article is available online.

## Acknowledgements

This work was supported by the Centre National pour la Recherche Scientifique (CNRS), the University Hospital of Montpellier (Centre Hospitalier Regional Universitaire, CHRU), and by the French National Research Agency (ANR Biopsy). A.C. was a recipient of a fellowship from the French Ministry of Health and a Resident at the University Hospital of Montpellier and is now supported by the Innovation Fellowship from the Washington Research Foundation and the Human Frontier Science Program. We thank Jerome Bonnet and Drew Endy for fruitful collaboration, insightful perspectives, and invaluable discussions. We thank Kathy Wei, Marc Lajoie, Anindya Roy, Scott Boyken, and Daniel Adriano Silva Manzano for helpful feedback during the preparation and revision of the manuscript.

## Author contributions

AC and FM designed the study. AC carried on computational biochemical circuit design, experimental construction, *in vitro* assays, and statistical analysis. AC, PA, and FF performed *in silico* experiments and carried out software development and computational work. ER collected clinical samples. All authors participated in the interpretation of the results and in the editing of the manuscript.

## Conflict of interest

The authors declare that they have no conflict of interest.

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
