## [Review Process File · Molecular Systems Biology]

Computer-aided biochemical programming of synthetic microreactors as diagnostic devices

Alexis Courbet, Patrick Amar, François Fages, Eric Renard & Franck Molina

Review timeline:

Submission date:	26 June 2017
Editorial Decision:	6 August 2017
Revision received:	22 November 2017
Editorial Decision:	22 December 2017
Revision received:	24 January 2018
Editorial Decision:	5 February 2018
Revision received:	26 February 2018
Accepted:	21 March 2018

Editor: Thomas Lemberger

Transaction Report:

1st Editorial Decision

6 August 2017

Thank you again for submitting your work to Molecular Systems Biology. We have now heard back from the three referees who agreed to evaluate your manuscript. As you will see from the reports below, the referees find the topic of your study of potential interest. They raise, however, several concerns on your work, which should be convincingly addressed in a major revision of this work.

Without repeating all the points made in the reports below, one of the major concerns raised by all three referees is the need of serious overhaul in the writing of the manuscript to clarify the text and significantly improve the rigor and depth of the reporting. It is of utmost importance that the methodologies, computational and experimental steps are reported with sufficient detail and clarity so that others can understand what has been done and how it has been done. As reviewer #2 summarizes it, the present "paper comes across as an implementation (or perhaps demonstration)" and it should be transformed into a rigorous scientific report that makes the conceptual and technical advances clear. We would therefore invite you to have an in-depth look at the current main text and revised it in these directions.

We would also encourage you also maximally use our ability to link source data or computer code to individual figures, to associate datasets/code with the manuscript. If necessary, use literate programming or executable notebooks (eg Jupyter) to describe computational pipelines. Please describe the experimental methods in Materials & Methods such that they are reproducible, including reagents, instrumentation, concentrations and initial conditions, and experimental steps. When necessary and appropriate use SBOL (<http://sbolstandard.org/>) to described circuits.

When you resubmit your manuscript, please download our CHECKLIST (<http://embopress.org/sites/default/files/Resources/EP_Author_Checklist_Master.xlsx>) and

include the completed form in your submission. *Please note* that the Author Checklist will be published alongside the paper as part of the transparent process
<<http://msb.embopress.org/authorguide#transparentprocess>>.

If you feel you can satisfactorily deal with these points and those listed by the referees, you may wish to submit a revised version of your manuscript. Please attach a covering letter giving details of the way in which you have handled each of the points raised by the referees. A revised manuscript will be once again subject to review and you probably understand that we can give you no guarantee at this stage that the eventual outcome will be favorable.

REVIEWER REPORTS

Reviewer #1:

Summary:

This manuscript demonstrates advances in the design of synthetic protosensors that perform biosensing and computing operations in vitro. Previously, such advances have been limited because the multidimensional search space for assembling biosynthetic programs is too large. To address this problem, the authors uniquely combine machine learning, microfluidics, and in vitro systems. Specifically, the authors demonstrate that they can use multiplexed sensors to implement a diagnostic algorithm to discriminate between pathological glycosuria patient clinical samples. This may be the first application of in vitro systems to the detection of disease states in clinical samples and therefore should be of general interest and importance. In general, the study is performed at a high level. However, the writing style is dense and tough to access and some of the figures are difficult to interpret as the text is too small. This should be addressed. While the manuscript is strong, I have some concerns. First, I wonder if some of the key conclusions/modules set forth by the authors have already been realized and reported. For example, the idea of synthetic vesicles with alpha-hemolysin transmembrane pores is not new. Second, I had a difficult time figuring out what optimization went into initial determination of the biochemical systems in each of the three individual protosensors. This should be made more clear. An additional concern was the selection of testing for a diabetes detector. The authors start with the idea that their "diagnostic algorithm discriminates between all acute complications of diabetes and achieves differential diagnosis." However, with clinical samples, they only used the GluONe protosensor. Despite these concerns, the manuscript seems well done. Additional concerns for consideration are listed below:

1. One concern is the choice of model detection system. It seems as though the focus on diabetes was in part motivated because of the accessibility of purified enzymes that could be purchased off the shelf. What was unclear was how this compares to existing strategies for detecting diabetes? What problem does this detector solve? This should be clearly articulated.
2. What is the possibility of making the enzyme system in the vesicle by in vitro translation? Have the authors tried this? I realize this adds an extra layer of complexity, but it would enhance impact.
3. The authors demonstrate stability of the protocells in Figure 4 by having a fluorophore attached to a large IgG protein. While this tests the stability of the protocell, it doesn't give an indication of how long enzymes would be stable. Can the authors show how stable their three protosensors (GluONe, LacOH, and GluNOx) are?
4. In Figure 5, the signal to noise of the GluNOx circuit in panel d does not seem too strong, as compared to the others. What strategies could be used to improve the validation of the "ON" signal? What happens if the reaction is left for more time? Is the product diffusing away?

Reviewer #2:

This paper describes a high level workflow for design and implementation of synthetic micro-reactors that can serve as diagnostic devices. The paper is interesting, though there are lots of little errors and issues that distract from the content (described in more detail below).

This paper has the challenge of describing a fairly complex workflow in a small number of pages

and I often found myself wanting to see more description about one of the components (eg HSIM or BIOCHAM). To some extent this is covered in the supplementary information, but this is disconnected from the main flow. As a consequence, the paper comes across as an implementation (or perhaps demonstration) of a lot of ideas that have been worked out elsewhere, rather than a wholly new piece of research. I don't have a constructive solution for this given the constraints of the journal format.

There are a number of modifications that I think would serve to improve the quality of the paper:

- * I found some of the terms in the abstract to be a bit too strong for what is presented. I question the terms "ultra-efficient" (perhaps efficient is enough; what does "ultra" mean?), "intractable" (I would agree with difficult, but intractable felt a bit strong). Some of these terms (like intractable) appear elsewhere and might be toned down a bit.

- * In the introduction, the authors claim that "to date no clear engineering principles or methodologies exist to design cell-free synthetic biochemical logic systems". I believe that Qian, Winfree, and Bruck's paper on see-saw gates (Nature, 2011) is a counter-example to this claim.

- * Last paragraph of subsection on computer aided biochemical programming: what was it interesting that the experiments should agree with the HSIM predictions? Isn't this the point of a predictive model? Perhaps explain a bit.

- * In the last paragraph before the Discussion section, there is a reference to Figure 7 but the manuscript does not contain a figure 7.

- * There are problems in reference numbering. References 63 and 64 appear twice in the reference list (the first two should be 60 and 61). Reference 70 is missing (the second reference #68 should be #70).

There were also many small errors that should be fixed:

- * Middle of the abstract, "microfluidic enabling" to "microfluidic technology enabling"? Seems like there is a missing word.

- * First paragraph of results: "allow signal transduction" should be "allowing signal transduction" to be in agreement with the other items in that list.

- * Second paragraph of results: "Proteins also offer the advantage to be mined and retrieved" needs to be reworded.

- * First paragraph of subsection on computer aided biochemical programming: Boolean Logic → Boolean logic.

(Many additional errata at this level of detail that should be picked up by a good copy editor.)

Reviewer #3:

In this paper the authors describe the development of membrane-encapsulated, enzyme-based biosensors (termed "protosensors"). For the development of the sensors, the enzyme circuits are generated automatically starting from truth tables and analysed extensively using computational methods. The sensors themselves are experimentally realized using DPPC liposomes based on double emulsions. The enzyme mix is encapsulated in liposomes created using microfluidics, whose membranes are made permeable with nanopores.

This is interesting work, but the text and argumentation is a little hard to follow. Parts of the article are quite lengthy and it is sometimes difficult to see what actually was done. It would be great if the authors could try to streamline the text (in particular Introduction and Discussion) and describe more clearly the specific work of this article early in the text.

Specific questions:

- could you describe the automatic generation of the enzyme circuits a little better in the main text (not only in the Supplement). As a user, what do you actually enter as an input to your software tool?
- in Figure 3 you mention "empirically determined concentration parameters". In general, how does your "rational" approach deal with variability of enzyme activities? Your circuits rely on commercially available enzymes, and you take parameters from the BRENDA database - but the parameters in this database vary widely as do enzymes from batch to batch. Put differently: how "rational" can such design be after all?
- it is truly remarkable that the DPPC vesicles appear stable over several months - but how stable are the enzymes over this time, and how is their activity affected by compartmentalization? Can you prepare the complete protosensors and store for a long time?
- among the things that remain unclear to the referee is the role of the alpha hemolysin nanopore and how it affects sensor performance. This is only mentioned in passing and a little in the Supplement. Related, the model includes the permeability of the protosensors, but this is never really discussed in the text.
- what is the role of osmotic pressure in the performance of the protosensors, and how do you deal with it?
- Figure 4c: technically, how can you perform confocal microscopy on the *same* vesicle over several months?
- Figure 5: the definition of "satisfaction degree" is unclear to this referee
- it is not clear to the referee whether you actually performed a multiplexed experiment with all three sensors in parallel, or not. Can you please clarify and state more clearly.

In the Supplement:

- why do you perform a stochastic analysis of the circuits? Is it expected to play a role for such large systems?
- in the permeability model - what is dx ? Is it the same as x ?

1st Revision - authors' response

22 November 2017

Detailed description of changes:

Reviewer #1:

Summary:

This manuscript demonstrates advances in the design of synthetic protosensors that perform biosensing and computing operations in vitro. Previously, such advances have been limited because the multidimensional search space for assembling biosynthetic programs is too large. To address this problem, the authors uniquely combine machine learning, microfluidics, and in vitro systems. Specifically, the authors demonstrate that they can use multiplexed sensors to implement a diagnostic algorithm to discriminate between pathological glycosuria patient clinical samples. This may be the first application of in vitro systems to the detection of disease states in clinical samples and therefore should be of general interest and importance. In general, the study is performed at a high level. However, the writing style is dense and tough to access and some of the figures are difficult to interpret as the text is too small. This should be addressed. While the manuscript is strong, I have some concerns.

We thank the reviewer for appreciating our work. We acknowledge and fully understand the reviewer's reserve on the ton and style of the manuscript. We have tried to bring the necessary modifications to the manuscript that we think bring more clarity.

First, I wonder if some of the key conclusions/modules set forth by the authors have already been realized and reported. For example, the idea of synthetic vesicles with alpha-hemolysin transmembrane pores is not new.

We agree that the approach based of phospholipid vesicles with hemolysin transmembrane pores is definitely not new. Indeed, multi-enzyme reaction in confined environments such as liposomes have been the subject of extensive studies: enzymatic reactions confined inside small volumes, fuel cells and enzymatic micro/nanoreactors. We therefore modified the text to highlight the interest and goals of the study, which is to our knowledge the first attempt to critically evaluate some of the necessary steps to automate the design of such systems that behave according to logic specifications.

Second, I had a difficult time figuring out what optimization went into initial determination of the biochemical systems in each of the three individual protosensors. This should be made more clear.

We understand and again agree with the reviewer, we have brought necessary modification to highlight the rationale and methodology behind the design of the biochemical systems.

An additional concern was the selection of testing for a diabetes detector. The authors start with the idea that their "diagnostic algorithm discriminates between all acute complications of diabetes and achieves differential diagnosis." However, with clinical samples, they only used the GluONE protosensor.

We acknowledge that this is indeed a limitation in our study. Having access to good quality clinical samples of non-stabilized diabetic patients with acute complications is extremely challenging, if not impossible. This is the reason why we limited our study to the use of the clinical samples that could be rigorously harvested, namely patient presenting pathological glycosuria. We modified our claims to indicate that these systems could in theory perform differential diagnosis, but has unfortunately not been evaluated in all circumstances.

Despite these concerns, the manuscript seems well done. Additional concerns for consideration are listed below:

1. One concern is the choice of model detection system. It seems as though the focus on diabetes was in part motivated because of the accessibility of purified enzymes that could be purchased off the shelf. What was unclear was how this compares to existing strategies for detecting diabetes? What problem does this detector solve? This should be clearly articulated.

We decided to target this model pathology for multiple reasons, which are not related to the accessibility of enzymes: (i) the real socio-economic burden of acute diabetes complications and the effective benefit of ready to use, smart, disposable systems, which would increase the potential impact of such a proof of concept (ii) the history of our lab, where we have developed over the years the human expertise, resources and collaborations with expert clinicians and specialist related to endocrinology and diabetes (iii) Our ultimate goal is to develop multiple approaches for the design of programmable biologics as logic biosensing devices, and to be able to compare different approaches we have decided to target the same pathology and biomarkers. However, the capability to purchase commercial enzymes is indeed an advantage to increase the reproducibility and easiness of experiments in the context of such a pilot study, which mostly focused on developing computational and microfluidic tools. We are however already extending the approach discussed in this study to different pathologies, that require a larger panel of enzyme. We agree that we did not discuss the potential place of the technology we develop here in the current gold-standard of diabetes diagnostics, we therefore added a discussion on the 6th paragraph of the discussion.

2. What is the possibility of making the enzyme system in the vesicle by *in vitro* translation? Have the authors tried this? I realize this adds an extra layer of complexity, but it would enhance impact.

This approach has been explored in the past with success. However, this methodology is probably not suited to achieve our goals, since the concentrations of enzyme that could be obtained using encapsulated *in vitro* translation systems in liposomes are very low and thus of no interest to our study. The concentrations that could in theory be obtained would not be sufficient, nor sufficiently and reliably controlled to allow the rational design of such biochemical logic systems according to temporal logic specifications. Indeed, as described in our study, encoding of kinetic, logic and robustness parameters necessitate precise control on high concentrations of enzymes. Moreover, the

molecular complexity of *in vitro* translation systems could certainly add considerable unreliability to the models of the biochemical systems, and would greatly reduce the capabilities of our model checking and parameter optimization approach. We however agree with the reviewer that this approach could be interesting in the future to expand this system and give them interesting properties for other applications.

3. The authors demonstrate stability of the protocells in Figure 4 by having a fluorophore attached to a large IgG protein. While this tests the stability of the protocell, it doesn't give an indication of how long enzymes would be stable. Can the authors show how stable their three protosensors (GluONe, LacOH, and GluNOx) are?

We thank the reviewer for raising that point, which was not clear in the current text, and we agree that the data we present does not give any indication on how long enzymes would remain stable in the protocells. This specific experiment was meant to test the actual stability and impermeability of the membrane, which is a prerequisite to achieve biochemical signal processing by preventing leaking of components in and out of the protocell. We did not address the biochemical stability of the enzymes as it did not seem of utmost relevance for this preliminary work and proof of concept, but we agree that this would be of considerable interest. However, many studies have explored this experimentally and showed that encapsulation of enzymes enhance their protection against denaturation and proteases (Nasseau M et al. *Biotechnol Bioeng.* 2001; Yoshimoto M. *Methods Mol Biol.* 2017; Yoshimoto M et al. *Biotechnol Prog.* 2008 ; Rodriguez-Nogales, *Journal of Chemical Technology & Biotechnology* , 2003). Therefore, we reasoned that enzyme stability should in theory be nearly equal or superior to the stability of the bulk enzymes in solution. We added a discussion of this point in the result and discussion sections to make it clearer.

4. In Figure 5, the signal to noise of the GluNOx circuit in panel d does not seem too strong, as compared to the others. What strategies could be used to improve the validation of the "ON" signal? What happens if the reaction is left for more time? Is the product diffusing away?

The data shown in figure 5 panel d was obtained using confocal microscopy. The fluorescence signal measured for each molecule after excitation with 488 (for DAF), and 561nm (DiI and Resorufin) lasers is not normalized, and therefore this data cannot be taken as a quantitative measure of signal. We argue that this is rather a qualitative measure of the on/off state of the output, and provides evidence of the presence of biochemical output molecules inside the protocells, which accounts for the measured signal in other experiments and proves that the biochemical machinery capable of producing the outputs are indeed encapsulated within the membrane and not in bulk solution. However, we were able to calculate signal-to-noise ratios for the data shown in Figure 3b, which are ~20, 34, 14, 26 dB for outputs 1, 2, 3 and 4 respectively.

Indeed, the reviewer is right to point out the effects of time in the measured output signal. Diffusion of the fluorescent output molecules out of the protocells should indeed be expected after some time. In this study, we decided to optimize our biochemical circuits with temporal logic specifications such that the maximal output would be obtained at 60 minutes after induction with input molecules, at which time we performed all the measurements presented in this study. We expect the systems to evolve toward thermodynamic equilibrium after the programmed biochemical operations are performed, and therefore to see the protocells gradually leak signal molecules in the surrounding solution. However, for the focus of the project we did not further investigate the evolution of these systems towards chemical equilibrium.

Reviewer #2:

This paper describes a high level workflow for design and implementation of synthetic micro-reactors that can serve as diagnostic devices. The paper is interesting, though there are lots of little errors and issues that distract from the content (described in more detail below).

This paper has the challenge of describing a fairly complex workflow in a small number of pages and I often found myself wanting to see more description about on of the components (eg HSIAM or BIOCHAM). To some extent this is covered in the supplementary information,

but this is disconnected from the main flow. As a consequence, the paper comes across as an implementation (or perhaps demonstration) of a lot of ideas that have been worked out elsewhere, rather than a wholly new piece of research. I don't have a constructive solution for this given the constraints of the journal format.

We thank the reviewer for appreciating our work and for providing very helpful recommendations, in this revision we have tried to bring the necessary corrections to all issues raised here.

We agree that our paper proposes a demonstration and combination of many previously described ideas. However, we argue here that it is the first realization of such a synthesis of many methodologies. Importantly, we also show the feasibility of such a high-level methodology by bringing novel computational tools and experimental solutions that we think are indeed necessary for realizing the programming of synthetic biochemical circuits. Microfluidics on the experimental end, and model checking on the computational side, are key technological support of such a workflow that we adapted here to our goals. Therefore, to our view our work is an original realization and proof-of-concept that could open up the way to the programming of biochemical micro-reactors.

There are a number of modifications that I think would serve to improve the quality of the paper:

*** I found some of the terms in the abstract to be a bit too strong for what is presented. I question the terms "ultra-efficient" (perhaps efficient is enough; what does "ultra" mean?), "intractable" (I would agree with difficult, but intractable felt a bit strong). Some of these terms (like intractable) appear elsewhere and might be toned down a bit.**

We thank the reviewer for raising this language issue and understand that the tone of the paper should be modified to sound more rigorous. We have brought the necessary modifications.

*** In the introduction, the authors claim that "to date no clear engineering principles or methodologies exist to design cell-free synthetic biochemical logic systems". I believe that Qian, Winfree, and Bruck's paper on see-saw gates (Nature, 2011) is a counter-example to this claim.**

We could not agree more with the reviewer, and apologize for this lack of precision in this sentence that leads to confusion. Strand displacement mechanisms have indeed been widely used to design complex biochemical logic circuits. However, we argue that no existing framework allow the use of arbitrary biomolecular building blocks of various nature such as metabolites, enzymes, nucleic acids and integrate them in comprehensive models that allow simulation and design. We modified the introduction to make clear that we referred here to reaction based biochemical systems composed of a variety of reactive species of different nature, and not only restricted to Watson-Crick base pairing programmability of DNA circuits.

*** Last paragraph of subsection on computer aided biochemical programming: what was it interesting that the experiments should good agreement with the HSIM predictions? Isn't this the point of a predictive model? Perhaps explain a bit.**

This is indeed the point of a predictive model and this expression is confusing. The important thing here is that this agreement between experimental data and simulations shows that HSIM models can capture relevant biochemical reactivities that condition the behavior of these systems. We therefore modified the text.

*** In the last paragraph before the Discussion section, there is a reference to Figure 7 but the manuscript does not contain a figure 7.**

*** There are problems in reference numbering. References 63 and 64 appear twice in the reference list (the first two should be 60 and 61). Reference 70 is missing (the second reference #68 should be #70).**

There were also many small errors that should be fixed:

* Middle of the abstract, "microfluidic enabling" to "microfluidic technology enabling"? Seems like there is a missing word.

* First paragraph of results: "allow signal transduction" should be "allowing signal transduction" to be in agreement with the other items in that list.

* Second paragraph of results: "Proteins also offer the advantage to be mined and retrieved" needs to be reworded.

* First paragraph of subsection on computer aided biochemical programming: Boolean Logic → Boolean logi.

We thank the reviewer for bringing these errors to our attention.

Reviewer #3:

In this paper the authors describe the development of membrane-encapsulated, enzyme-based biosensors (termed "protosensors"). For the development of the sensors, the enzyme circuits are generated automatically starting from truth tables and analysed extensively using computational methods. The sensors themselves are experimentally realized using DPPC liposomes based on double emulsions. The enzyme mix is encapsulated in liposomes created using microfluidics, whose membranes are made permeable with nanopores.

This is interesting work, but the text and argumentation is a little hard to follow. Parts of the article are quite lengthy and it is sometimes difficult to see what actually was done. It would be great if the authors could try to streamline the text (in particular Introduction and Discussion) and describe more clearly the specific work of this article early in the text.

We thank the reviewer for appreciating our work and for giving it careful consideration. We acknowledge and fully understand the reviewer's reserve on the ton and style of parts of manuscript, which we have tried to address in this revision.

Specific questions:

- could you describe the automatic generation of the enzyme circuits a little better in the main text (not only in the Supplement). As a user, what do you actually enter as an input to your software tool?

As judiciously recommended we added more details on the automated generation of synthetic enzyme circuits in the main text in the first result section to make clear what inputs have to be fed in our software tool.

- in Figure 3 you mention "empirically determined concentration parameters". In general, how does your "rational" approach deal with variability of enzyme activities? Your circuits rely on commercially available enzymes, and you take parameters from the BRENDA database - but the parameters in this database vary widely as do enzymes from batch to batch. Put differently: how "rational" can such design be after all?

We agree with the reviewer that the data on kinetic parameters present in the BRENDA database may in certain case not allow for rational design of such circuit or are likely to be a limiting factor. Data on enzymatic parameters or reactivity may be lacking for certain enzymes or non-relevant in context. However, the fact that we obtain HSIM trajectories in agreement with the data we collected is a good indication that BRENDA information can be trusted at least for common, robust and widely used enzyme in common biochemistry buffers. For instance, BRENDA entries for the enzymes used in this study (Glucose 1-Dehydrogenase EC 1.1.1.47, Alcohol Dehydrogenase EC 1.1.1.1, Horseradish peroxidase 1.11.1.7, Lactate Oxidase 1.13.12.4, Nitrate Reductase EC 1.7.1.1) have consistent species, buffer, condition and substrate specific kinetic parameters. In our study,

after comparing experimental kinetic data we measured and HSIM simulation we discovered that the kinetic parameters obtained in the BRENDA database were capable of producing the behavior observed. We agree that this may however not be the case with other enzymes, or that variability in the enzyme batch could forbid the use of kinetic parameters extracted from this database. However, it is possible to overcome these possible issues by measuring experimentally kinetic parameters of enzyme in context, and then use this data to build models and then perform model checking to optimize temporal logic specifications.

- it is truly remarkable that the DPPC vesicles appear stable over several months - but how stable are the enzymes over this time, and how is their activity affected by compartmentalization? Can you prepare the complete protosensors and store for a long time?

We thank the reviewer for raising that point, which was also raised by the first reviewer. It was not clear enough in the current text, and we agree that the data we present does not give any indication on how long enzymes would remain stable in the protocells. This specific experiment presented in Figure 4 was meant to test the stability and impermeability of the membrane, which is a prerequisite to achieve biochemical signal processing by preventing the leaking of components in and out of the protocell. This data is in accordance with previous report on the stability of DPPC vesicles (Armengol et al., 1994, Journal of Microencapsulation; Park et al., 2010, Journal of Colloid and Interface Science; Phospholipids Handbook, Gregor, 1993). We did not address the biochemical stability of the enzymes nor the effect of compartmentalization on their stability as it did not seem of utmost relevance for this preliminary work and proof of concept, but we agree that this would be of considerable interest. In this study, we prepared batches of protosensors in Buffer A (10% v/v methanol, 15% w/v glycerol, 3% w/v pluronic F68 in PBS) in which they were stored at 4C before being used within a week of fabrication to collect the data presented. As it was not the primary focus of the study we did not further investigate the maximum biochemical stability of these systems, but again, this would be of considerable interest in future experiments. Besides, many studies have explored this experimentally and showed that encapsulation of enzymes enhance their protection against denaturation and proteases (Nasseau M et al. Biotechnol Bioeng. 2001; Yoshimoto M. Methods Mol Biol. 2017; Yoshimoto M et al. Biotechnol Prog. 2008; Rodriguez-Nogales, Journal of Chemical Technology & Biotechnology, 2003). Therefore, we reasoned that enzyme stability should in theory be nearly equal or superior to the stability of the bulk enzymes in solution. We added a discussion of this point in the result and discussion sections to make it clearer.

- among the things that remain unclear to the referee is the role of the alpha hemolysin nanopore and how it affects sensor performance. This is only mentioned in passing and a little in the Supplement. Related, the model includes the permeability of the protosensors, but this is never really discussed in the text.

We agree that a more detailed discussion of the role of alpha hemolysin and permeability is needed in the main text, and we therefore modified the second and fourth section of the results accordingly. Activation of the sensor's biochemical circuits by the input is completely reliant on the selective permeation of small molecules, which is made possible by the pore forming assembly of alpha hemolysin channels that are introduced at $t=0$. Then, the diffusion of small molecules is taken into accounts in our mathematical models by integrating Fick's equations of diffusion and accessible permeability coefficients that are known. Larger molecules, such as enzymes however, are prevented from leaking out because of their size. The concentration of hemolysin is expected to play a major role in the protosensors response, since it would have an impact on the diffusion kinetics and therefore on the kinetics of the whole protosensors behavior. Depending on the concentration of hemolysin chosen by the experimenter, models have to be modified and the biochemical circuits have to be optimized accordingly to guaranty a correct behavior. Regardless of the concentration of hemolysin chosen, we expect that a specific behavior can be implemented by using Biocham to compute the optimal concentration of enzyme that satisfy chosen temporal logic specifications.

- what is the role of osmotic pressure in the performance of the protosensors, and how do you deal with it?

The reviewer is here raising a very interesting point that will receive further scrutiny in future work, as we decided not investigate the consequences of osmotic pressure in this study. Although the physic based models of diffusion incorporated in HSIM are in theory capable and should support the

description of the evolution of solute concentration in and out of the membrane due to osmotic pressure, exposing photosensors to dramatically high or low osmotic pressure environments could indeed have an important impact on the membrane stability and capacity to insulate the circuit from the surrounding. In the context of phospholipids bilayers used in this study, we envision that the most important issue caused by osmotic pressure is the threat posed to membrane integrity. Nevertheless, as further detailed in the discussion section, although they could prove useful for specific applications, we see the use of phospholipidic membranes as preliminary, proof-of-concept and non-optimal support for most application that would potentially necessitate increased osmotic pressure resistance. Phospholipidic membrane proved extremely convenient as model of study of microreactors, that can be easily generated, modeled and analyzed. We added more detailed discussion on this point in the discussion section to make clear that is point is of great importance and that we are aware that further will be required in that direction.

- Figure 4c: technically, how can you perform confocal microscopy on the *same* vesicle over several months?

We did not track the same protocell over the course of three month, as the author is judiciously suspecting. In this figure, we sought to demonstrate the capacity of the membrane to encapsulate a biochemical cargo over time without leakage. Each protocell having the same size and same concentration of biochemical species content as ensured by the microfluidic generation method, we can see in Figure 4c that the fluorescence intensity of the inner medium does not decrease over time, indicating that the biochemical content does not leak through the membrane.

- Figure 5: the definition of "satisfaction degree" is unclear to this referee

We thank the reviewer for pointing out that a description of satisfaction degree is lacking in this version of the text. We therefore added a succinct definition of this concept in the fourth result section, which originates from previous work cited in the study (Rizk et al., Proceedings of the fourth international conference on Computational Methods in Systems Biology, volume 5307 of Lecture Notes in Computer Science, pages 251-268. Springer-Verlag, October 2008; Rizk et al., Bioinformatics. 2009 Jun 15; 25(12): i169–i178.) Briefly, Rizk et al. showed how a continuous satisfaction measure of LTL(R) Formulae can be computed to serve as a fitness function in order to find biochemical kinetic parameter values satisfying a set of biological properties formalized in temporal logic, in order to evaluate numerically the adequateness of a model relative to temporal logic specifications, which provides a quantitative notion of robustness. The satisfaction degree is normalized such that it ranges between 0 and 1, with a satisfaction degree equal to 1 when the property is true and tending toward 0 when the system is far from satisfying the expected property.

- it is not clear to the referee whether you actually performed a multiplexed experiment with all three sensors in parallel, or not. Can you please clarify and state more clearly.

We apologize and agree with the reviewer on the lack of clarity on this matter, we modified the section four of results to make clear that we indeed perform multiplexed experiments with the three sensors at the same time in the same sample. This experiment was of considerable interest because it shows that the use of membrane encapsulation of synthetic biochemical circuits provides capabilities analogous to insulations in electronic circuits. This way multiple circuits can operate simultaneously without molecular “short-circuits”. These properties could open the way to the construction and standardization of circuits elements that could be put together in a modular way, potentially allowing the construction of more complex systems.

In the Supplement:

- why do you perform a stochastic analysis of the circuits? Is it expected to play a role for such large systems?

In our protocol for modeling biochemical circuits, in a first step we indeed use stochastic simulations for the reason that they potentially provide higher reliability and precision compared to ODEs without risking potential aberrations characteristic to ODEs simulations (i.e. fractional concentrations). At this point this is considerably advantageous because still computationally inexpensive, there would be no advantage to use ODEs modelling in this context.

- in the permeability model - what is dx? Is it the same as x?

This could indeed have been unclear and we added another mention of the definition of x in this section. Since the movement of diffusing molecules depends on the concentration gradient, the rate of diffusion is directly proportional to the concentration gradient (dC/dx) across the membrane. The concentration gradient, dC/dx , is the difference in molecule concentration inside and outside of the cell across a protocell membrane of width dx . This is equivalent to $(C_{out} - C_{in})/dx$ where C_{out} and C_{in} are the substrate concentrations inside and outside the protocell, and dx is the width of the cell membrane. When the concentration outside the cell (C_{out}) is larger than inside the cell (C_{in}), the concentration gradient (dC/dx) will be positive, and net movement will be into the cell (positive value of dn/dt).

2nd Editorial Decision

22 December 2017

Thank you again for submitting your work to Molecular Systems Biology. We are now globally satisfied with the modifications made and I am pleased to inform you that we will be able to accept your manuscript for publication in Molecular Systems Biology pending the following points:

- in the Results and Materials & Methods sections, reference is made to Silicell Maker, NetGate and NetBuild. We were unable to find these software and it remains unclear what are the input/output and what computation is been made at this step of the design. What are the 'natural networks' being mined and what does 'mining' mean in this context? The reference 70 does not seem to refer to a formal publication and should as such should be removed. We would therefore kindly ask you to clarify this aspect of the analysis in the main manuscript and provide direct links to these software or alternatively Docker images to enable researchers to reproduce the analysis.

- the Results section reports that HSIM was refined. It is not clear what is the nature of these refinements and where the updated HSIM simulator can be found.

- please include in line the full URL to all the software platforms used in this study (see also above).

- please provide all the code files, including sbml as text-only files and NOT as PDF. Code files should be named "Computer Code EV[n]" and should be called out explicitly from the main text. If several files cover the same process (eg a notebook file and a batch file), they can be zipped in a single archive for clarity. Code files should be zipped together with a plain text README file providing a description of the content of the archive and instructions on how to use the files. Zipping a code file is also a simple way to 'protect' its extension (.cb or .xml or .sbml, etc...).

- we note that the checklist is practically empty. No indication of which statistics have been used seem to appear in the figures nor in the checklist. Please complete this information where applicable or indicate 'na' in the checklist. The list of software used should also be listed with the relevant links

- ORCID identifiers are required for all corresponding authors

- Please add a running title and up to five keywords on the first or second page of the manuscript.

- Please update the references and callouts from numerical to alphabetical to match the MSB reference style. <http://msb.embopress.org/authorguide#referencesformat>

- Please upload the main figures as individual figure files (high resolution). Please remove the figure legends from the figures. The figure legends should only appear once in the manuscript file.

- Please rename Movie 3 to Movie EV1 and update the callout in the manuscript accordingly.

- Please zip the movie with a movie legend txt or doc file.

- For the APPENDIX:

- Please rename the file called Supplemental Materials -> Appendix.

- Please rename the Appendix tables and figures to Appendix Table S[n] and Appendix Figure S[n].

- The Appendix Supplementary reference list should also follow the MSB reference style.
- Please mention Appendix Figure S8C in the legend.
- Please rename figure S25 to Appendix figure S15.
- Please specify for which circuit the example of BIOCHAM code file is provided
- Please add an explicit callout to figure 6B in the main text
- Callouts to the EV Movie, the Appendix figures and tables need to be updated to match our nomenclature style.

2nd Revision - authors' response

24 January 2018

Detailed amendments and responses to each point

-In the Results and Materials & Methods sections, reference is made to Silicell Maker, NetGate and NetBuild. We were unable to find these software and it remains unclear what are the input/output and what computation is been made at this step of the design. What are the 'natural networks' being mined and what does 'mining' mean in this context? The reference 70 does not seem to refer to a formal publication and should as such should be removed. We would therefore kindly ask you to clarify this aspect of the analysis in the main manuscript and provide direct links to these softwares or alternatively Docker images to enable researchers to reproduce the analysis.

We apologize for omitting to feature a link to further references, documentation, and access to software executables in question. We added to Material and Methods a link to a website where all this resource can be found. We modified the reference so that it would properly refer to the specific conference publication. We further modified the Results section of the main text to clarify that *mining* in this context refers to the automated implementation of a formal logic function by a set of biochemical reactions extracted from natural networks using the software NetGate. Additionally, *natural networks* refer to curated metabolic networks from multiple organisms and domains of life, such as the ones that can be found in MetaCyc Metabolic Pathway Database. In this context we define mining as the automated implementation of a formal logic function by a set of biochemical reactive species extracted from natural metabolic networks. Since we aim at programming biochemical logic circuits using multiple reactions taking place simultaneously in a microreactor, we reason that mining for molecular species within the same metabolic context *in vivo* would minimize possible failure modes. NetGate defines biochemical logic gates by their truth table, the set of molecular species representing inputs, output, and enzyme. NetGate takes as inputs (i) a SBML file describing an input metabolic network and (ii) a list of truth tables corresponding to the logic gates that are to be searched in the metabolic network. First, all the possible implementations of the logic gates are enumerated; Then, these implementations are checked against the given list of truth tables and the gates found are sorted and output. The gates implementations are searched in subnetworks extracted from the original metabolic network. These subnetworks are built starting from one reaction of the original network, the seed, then adding successively other reactions that are tied to this seed. To get all the subnetworks, this process is repeated starting from all the reactions of the original metabolic network. Then, for each of the given gate description, all the possible implementations are searched within each subnetwork. All the mappings of the inputs of the gate to the inputs of the subnetwork are successively checked to see if all the lines of truth table of the gate description can be obtained. The in-depth description of the algorithm behind NetGate can be found in Bouffard et al. Additionally, we added some details about NetGate in SI.

- The Results section reports that HSIM was refined. It is not clear what is the nature of these refinements and where the updated HSIM simulator can be found.

We thank the editor for bringing this lack of clarity to our attention. We modified the *In silico design and simulation* Methods section to feature the detail of recent improvements that we brought to HSIM for the purpose of this study, which consist of: multisubstrate enzymatic mechanisms implementation (Ordered sequential bi-bi mechanism, Ping Pong bi-bi mechanism, and random sequential bi-bi mechanism), automated translation of probability from HSIM stochastic simulator to mass action rates for BIOCHAM ODE solver, the implementation of physical models

for protosensor membrane permeability, as well as SBML support. The latest HSIM executables can be found and downloaded from Patrick Amar's website where a URL link to it can be found in material and methods.

- please include in line the full URL to all the software platforms used in this study (see also above).

We added to the Methods section in line URL links to all the software used in this study, which are:
[Silicell Maker software suite](https://silicellmaker.lri.fr/) <https://silicellmaker.lri.fr/>
[HSIM](https://www.lri.fr/~pa/Hsim/) <https://www.lri.fr/~pa/Hsim/>
[BIOCHAM](https://lifeware.inria.fr/biocham/) <https://lifeware.inria.fr/biocham/>
[BIOCHAM online server](http://lifeware.inria.fr/biocham/online/) <http://lifeware.inria.fr/biocham/online/>

- please provide all the code files, including sbml as text-only files and NOT as PDF. Code files should be named "Computer Code EV[n]" and should be called out explicitly from the main text. If several files cover the same process (e.g. a notebook file and a batch file), they can be zipped in a single archive for clarity. Code files should be zipped together with a plain text README file providing a description of the content of the archive and instructions on how to use the files. Zipping a code file is also a simple way to 'protect' its extension (.cb or .xml or .sbml, etc...).

We apologize for this issue, during the last submission we resolved to submit PDF files of the code for the reason that an apparent technical issue on the MSB website prevented from uploading archives. Hoping this is now fixed, we will try again to upload a single archive file containing the files along with a README as advised. Additionally, we have modified the main text of the manuscript to explicitly refer to these files as properly named "Computer Code EV[n]".

- we note that the checklist is practically empty. No indication of which statistics have been used seem to appear in the figures nor in the checklist. Please complete this information where applicable or indicate 'na' in the checklist. The list of software used should also be listed with the relevant links

We brought the necessary modification to figure legends as well as the checklist.

- ORCID identifiers are required for all corresponding authors
- Please add a running title and up to five keywords on the first or second page of the manuscript.

The following running title and five keywords were added on the first page of the manuscript:

Running title: Biochemically programmed microreactors

Keywords: synthetic biochemical logic circuits, biochemical programming, computational biochemical circuit design, synthetic microreactors, diagnostics

- Please update the references and callouts from numerical to alphabetical to match the MSB reference style. <http://msb.embopress.org/authorguide#referencesformat>

- Please upload the main figures as individual figure files (high resolution). Please remove the figure legends from the figures. The figure legends should only appear once in the manuscript file.

- Please rename Movie 3 to Movie EV1 and update the callout in the manuscript accordingly.

- Please zip the movie with a movie legend txt or doc file.

- For the APPENDIX:

- Please rename the file called Supplemental Materials -> Appendix.

- Please rename the Appendix tables and figures to Appendix Table S[n] and Appendix Figure S[n].

- **The Appendix Supplementary reference list should also follow the MSB reference style.**
- **Please mention Appendix Figure S8C in the legend.**
- **Please rename figure S25 to Appendix figure S15.**

- **Please specify for which circuit the example of BIOCHAM code file is provided**

- **Please add an explicit callout to figure 6B in the main text**

- **Callouts to the EV Movie, the Appendix figures and tables need to be updated to match our nomenclature style.**

We thank the editor for pointing us to all of these issues. We addressed all of these points and brought the necessary modifications to the manuscript.

3rd Editorial Decision

5 February 2018

Thank you again for submitting your work to Molecular Systems Biology.

I am afraid that we still have issues with regard to the points we raised in the previous round. We would be grateful if you could carefully consider the issues below since they keep delaying publication.

Please note that we will NOT be able to grant additional rounds of revisions and we would therefore need these issues to be definitively settled. If necessary, we are happy to provide further guidance over the phone.

#NetGate

We thank you for adding more details on the NetGate software platform and its principle. We could (we some difficulty) find the text of the Conference proceedings where the platform was discussed (<https://assb.lri.fr/Proceedings/LivreStrasbourg-15.pdf>). We understand that NetGate takes as input SBML files describing metabolic networks and finds possible solutions for biochemical implementations that correspond to the truth table of the logic gate of interest.

It remains however unclear how NetGate was used in the context of this specific study.

While the text mentions that NetGate takes as input metabolic networks "such as the one of Appendix Figure S10A, but can consist of significantly larger networks such as the ones that can be found in the MetaCyc Metabolic Pathway Database", we could not find which specific set of networks have actually been used in the context of this study. Did you use MetaCyc, KEGG, the whole database, a subset of it or some selected networks? What was exactly the input to NetGate in this study, how many networks, from what source/organisms?

In addition, it is also unclear what was the output. The main text states: "in silico solutions for biochemical implementations of protosensors that could execute the particular diagnostic algorithm for acute diabetes specification were found". It is unclear whether the few examples shown are the only solutions found by NetGate, or whether they were picked from a larger set of possible solutions and if yes, from how many and how.

In summary, these steps and results should be reported clearly in a way that allows other to reproduce the analysis.

Statistics

With regard to the checklist, we appreciate the following row was filled:

"5. For every figure, are statistical tests justified as appropriate? Yes"

However, it seems that none of the figures include any statistical tests except Fig 6, where the only P

value of the entire article is reported. Even in this case it is not clear what test was applied ("the data was processed using the software SigmaPlot"). We do not want to impose statistical tests where they are not informative, but would need you to address this point seriously.

3rd Revision - authors' response

26 February 2018

Detailed amendments and responses to each point

#NetGate

We thank you for adding more details on the NetGate software platform and its principle. We could (we some difficulty) find the text of the Conference proceedings where the platform was discussed (<https://assb.lri.fr/Proceedings/LivreStrasbourg-15.pdf>). We understand that NetGate takes as input SBML files describing metabolic networks and finds possible solutions for biochemical implementations that correspond to the truth table of the logic gate of interest.

It remains however unclear how NetGate was used in the context of this specific study.

While the text mentions that NetGate takes as input metabolic networks "such as the one of Appendix Figure S10A, but can consist of significantly larger networks such as the ones that can be found in the MetaCyc Metabolic Pathway Database", we could not find which specific set of networks have actually been used in the context of this study. Did you use MetaCyc, KEGG, the whole database, a subset of it or some selected networks? What was exactly the input to NetGate in this study, how many networks, from what source/organisms?

In addition, it is also unclear what was the output. The main text states: "in silico solutions for biochemical implementations of protosensors that could execute the particular diagnostic algorithm for acute diabetes specification were found". It is unclear whether the few examples shown are the only solutions found by NetGate, or whether they were picked from a larger set of possible solutions and if yes, from how many and how.

In summary, these steps and results should be reported clearly in a way that allows other to reproduce the analysis.

We thank the editor for stressing the lack of details on these two points, we agree that it is crucial to document this part of our work with clarity so that other researchers can benefit from it. Figure S1 of the Appendix was initially designed to address these points but was lacking information in that form, therefore we modified it to incorporate all the methodology surrounding the use of Silicell Maker programs NetGate and NetBuild. We also modified the main text to incorporate this important information.

To generate the synthetic biochemical circuits described in this study, we performed an organism agnostic search of all the sets of natural biochemical networks with overlapping enzymes, substrates or products related to the inputs and outputs of the different biochemical circuit we aimed at programming. The BRENDA database was queried for biochemical network containing enzymes, substrate or product related to glucose, acetone, NADH, resorufin, lactate, ethanol, and NOx metabolism (*i.e.* acetate fermentation, glycolysis, propanol degradation, disaccharide metabolism, ethanol fermentation, lactate fermentation, L-lactaldehyde degradation, methane metabolism, non-pathway related, NAD metabolism, Entner Doudoroff pathway, Nitrate assimilation, as well as non-pathway related peroxidase catalyzed reactions) and SBML files of these networks were downloaded from the BRENDA web interface (https://www.brenda-enzymes.org/search_result.php?a=200). We then used Silicell Maker to combine them into one large SBML network via the 'share clones' command. This large network (Computer Code EV1) was then used as input and mined using the program NetGate to identify all biochemical logic gates of ≤ 2 reactions ('Formula -> Extract Gates' command). In this first step, all the possible implementations of logic gates present in the input network are enumerated, and 775 logic gates were identified in this case. An implementation of a logic gate is a subnetwork where appropriate

biomolecular inputs and output are identified. The program NetBuild was then used ('Formula -> Enter Formula...' command) to find specific biochemical implementation corresponding to user-defined Boolean logic specifications. In this second step, all enumerated implementations are checked against the given truth tables and the gates found are sorted. The algorithm found unique implementations satisfying GluONE, LacOH and GluNOx biochemical logic from the input network with ≤ 2 reactions/gate. Increasing the number of reactions/gates would augment the solution space by yielding more implementations of higher complexity, but would be considerably more computationally expensive. To increase robustness and ease experimental and computational work for the purpose of this study, we pursued the smallest reactions set capable of recapitulating a specific logic formula.

We added the SBML file (Computer Code EV1) corresponding to the input network in the Appendix, and we updated Figure S1 with the input network and output circuit solutions found after running NetGate for GluONE, LacOH and GluNOx logic specifications. We made clear that from the input network described here and after specifying ≤ 2 reaction/gates, we found unique solutions for GluONE, LacOH and GluNOx.

Statistics

With regard to the checklist, we appreciate the following row was filled:

"5. For every figure, are statistical tests justified as appropriate? Yes"

However, it seems that none of the figures include any statistical tests except Fig 6, where the only P value of the entire article is reported. Even in this case it is not clear what test was applied ("the data was processed using the software SigmaPlot"). We do not want to impose statistical tests where they are not informative, but would need you to address this point seriously.

We thank the editor for pointing out this issue and understand that it is not satisfying in this present form. We therefore completed the checklist and brought the following improvements to the manuscript:

-For all bar plots (Figure 3 and Figure 5), we computed the P-value relative to induced vs non-induced conditions, according to a two-sided unpaired t-test and reported this information in the figure legends.

-In Figure 6, the ROC curve analysis is reporting a P value that tests the null hypothesis that the area under the curve really equals 0.50. 95% confidence interval of the AUC and the P value determines if the area value is significantly different from 0.5. The latter is the null hypothesis that we tested using the Mann-Whitney-Wilcoxon test and for which we reported the P-value.

-Figure 5: We reported that the flow cytometry data collection was carried on 10000 individual events.

-Concerning normality assumption: for ROC analysis, we used a Mann-Whitney-Wilcoxon test which does not require the assumption of normal distribution. For the other statistics, when two-sided unpaired t-test were used, we checked normality using the data analysis toolpack in Excel which provide the value of skewness and kurtosis from the descriptive statistic analysis as well as by inspecting graphically the degree of non-normality.

Corresponding Author Name: Alexis Courbet

Manuscript Number: MSB-17-7845